# Charge injection engineering at organic/inorganic heterointerfaces for high-efficiency and fast-response perovskite light-emitting diodes

Zhenchao Li[1,2,12], Ziming Chen [1,3,12] ✉, Zhangsheng Shi [4], Guangruixing Zou[5], Linghao Chu[1], Xian-Kai Chen [4,5,6,7,8] ✉, Chujun Zhang[9], Shu Kong So[9] & Hin-Lap Yip [1,5,10,11] ✉

The development of advanced perovskite emitters has considerably improved the performance of perovskite light-emitting diodes (LEDs). However, the further development of perovskite LEDs requires ideal device electrical properties, which strongly depend on its interfaces. In perovskite LEDs with conventional p-i-n structures, hole injection is generally less efficient than electron injection, causing charge imbalance. Furthermore, the popular hole injection structure of $NiO_x$/poly(9-vinylcarbazole) suffers from several issues, such as weak interfacial adhesion, high interfacial trap density and mismatched energy levels. In this work, we insert a self-assembled monolayer of [2-(9H-carbazol-9-yl)ethyl]phosphonic acid between the $NiO_x$ and poly(9-vinylcarbazole) layers to overcome these challenges at the organic/inorganic heterointerfaces by establishing a robust interface, passivating interfacial trap states and aligning the energy levels. We successfully demonstrate blue (emission at 493 nm) and green (emission at 515 nm) devices with external quantum efficiencies of 14.5% and 26.0%, respectively. More importantly, the self-assembled monolayer also gives rise to devices with much faster response speeds by reducing interfacial capacitance and resistance. Our results pave the way for developing more efficient and brighter perovskite LEDs with quick response, widening their potential application scope.

Metal-halide perovskites are emerging semiconducting materials with promising optical and electrical properties for light-emitting and other optoelectronic applications. Their organic-inorganic hybrid nature endows them with excellent flexibility in solution processing and dimensional modulation via tailoring the organic cations. Notably, the compositional and dimensional engineering of quasi-two-dimensional (quasi-2D) perovskites provides a feasible route to modulating their bandgaps and electronic structures[1]. These properties offer much room to improve perovskite light-emitting diodes (PeLEDs) from the material engineering level, leading to > 20% external quantum efficiencies (EQEs) for infrared, red, and green devices, as well as > 17% EQEs for blue and > 12% EQEs for white devices[1–6]. In addition to the hybrid nature of perovskite emitters, in terms of a whole device, it is also constructed with hybrid organic and inorganic layers, such as charge transport/injection layers that usually involve multilayers with inorganic/organic heterointerfaces. In conventional inorganic LEDs, the component layers are usually formed by epitaxial growth and connected via robust covalent bonds, forming inorganic/inorganic

heterointerfaces that are structurally well-defined. However, in PeLEDs, the random configuration of the inorganic/organic hetero-interface often leads to reduced physical robustness, imperfect hole injection, and undesired trap states, resulting in heterointerfaces with incommensurate nature.

An example of the inorganic/organic heterointerface is the $NiO_x$/PVK bilayer, which is widely used in PeLEDs for hole injection[7,8]. The interaction between PVK and $NiO_x$ is typically based on weak van der Waals forces[9], providing insignificant interfacial adhesion and surface passivation effects. In addition, considering that the hole transport property of PVK is mainly governed by the aromatic carbazole groups instead of the saturated polymer backbones, the random molecular interactions between amorphous PVK and $NiO_x$ likely reduce the effective contact for hole transport from $NiO_x$ to PVK[10]. Therefore, if these issues at the inorganic/organic heterointerface can be overcome, the hole injection property and the stability of the devices should be improved, benefiting PeLEDs with enhanced performance.

Here, to construct an improved $NiO_x$/PVK interface for PeLEDs, we introduced a self-assembled monolayer (SAM) of [2-(9H-carbazol-9-yl) ethyl]phosphonic acid (2PACz) as a bridge to form a $NiO_x$/2PACz/PVK tri-layer hole transport/injection layer. This SAM not only improved the robustness and energy level coupling between the $NiO_x$ and PVK layers, but also provided a much-improved surface trap passivation effect to the $NiO_x$ simultaneously[11–13]. Consequently, the turn-on voltages ($V_T$) of the blue and green PeLEDs were markedly reduced from 3.8 V to 2.1 V and from 2.7 V to 2.1 V, respectively. The resulting PeLEDs constitute one of the highest-performing blue PeLEDs [electroluminescence (EL) peak = 493 nm, EQE = 14.5%, luminance = 10,392 cd m$^{-2}$] and green PeLEDs (EL peak = 515 nm, EQE = 26.0%, luminance = 83,561 cd m$^{-2}$). Moreover, introducing the SAM accelerated the resistor-capacitor step response, contributing to a faster PeLED response speed. This further addresses one of the critical issues of PeLEDs to enable their potential use in optical communication applications such as light fidelity.

## Results

### Changes in the surface nature of the charge injection layer

SAMs with functional organic terminal groups and carboxyl or phosphonic acid end groups are widely used to modify the energy levels and surface energy of metal oxides, as well as passivate their surface traps[14–16]. Here, 2PACz (structure shown in Fig. 1a) was selected as the

SAM for two main reasons: (i) The phosphonic acid groups in 2PACz can form robust tridentate bonds with the $NiO_x$ surface[17], as illustrated in Fig. 1a, which is also confirmed by the X-ray photoelectron spectroscopy (XPS) studies (Supplementary Fig. 1); (ii) Owing to the high structural identity, the carbazole terminal groups in the 2PACz SAM should strongly interact with the carbazole units in PVK, resulting in dense packing that potentially facilitates augmenting charge injection and interfacial stability[18].

The formation of the 2PACz SAM on $NiO_x$ surface was further revealed by the contact angle measurements and Kelvin probe force microscopy (KPFM). Figure 1b, c show that small and large contact angles of approximately 21° and 104° were found for the $NiO_x$ and $NiO_x$/SAM surfaces, respectively. This suggests the hydrophilic hydroxylated $NiO_x$ surface had converted to a hydrophobic surface terminated with the carbazole groups. KPFM study also reveals that the surface potential of 0.588 V (corresponding to a work function of 3.832 eV) of the pristine $NiO_x$ had dramatically decreased to 0.185 V (corresponding to a work function of 4.235 eV) when the SAM was formed, suggesting an increase of work function by ~0.4 eV for the $NiO_x$/SAM sample. Moreover, the potential mappings in both cases show that the films were homogeneous, reflecting that 2PACz was uniformly bonded onto the $NiO_x$ surface with high coverage, which is also confirmed by the surface potential distribution results (Supplementary Fig. 2). Their morphologies were also characterized by scanning electron microscopy (SEM) and atomic force microscopy (AFM). The results shown in Supplementary Fig. 3 illustrate that the deposition of the 2PACz SAM had a negligible impact on the film morphology and roughness (~3.7 nm in both cases) and was thus amenable to the deposition of the upper functional layer.

### Adhesion of the PVK layer and its impact

PVK is widely used as a hole injection layer for blue and green PeLEDs due to its deep HOMO level[8,19]. Moreover, it is usually considered indissoluble in polar solvents such as dimethyl sulfoxide (DMSO), facilitating the processing of the perovskite precursor solution. Surprisingly, we found that PVK had considerable solubility in DMSO, as illustrated in the solubility tests in Fig. 2a. This finding triggered us to investigate the adhesion of the PVK layer on the $NiO_x$ surface, as poor physical adhesion might cause the PVK to dissolve more easily when processing the perovskite layer.

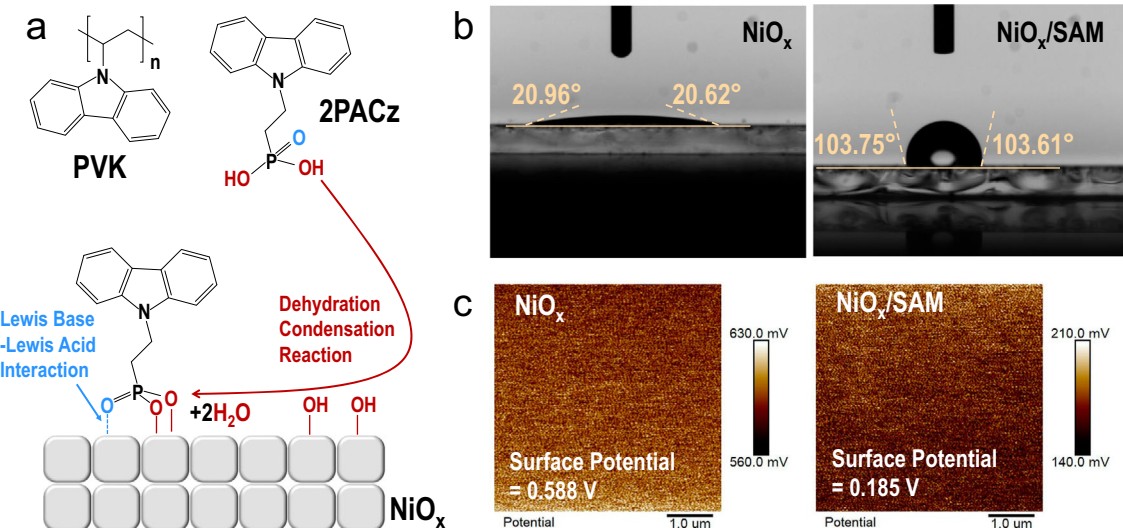

**Fig. 1 | Changes in the surface after the formation of the 2PACz SAM. a** The molecular structures of PVK and 2PACz, and a schematic diagram showing how the 2PACz molecules are bonded on the $NiO_x$ surface via tridentate bonds, i.e., one coordination bond (via Lewis base–Lewis acid interaction with the Ni atom) and two covalent bonds (via dehydration condensation reactions with the hydroxyl groups)[17]. **b** Contact angle measurements of $NiO_x$ and $NiO_x$/SAM films. **c** Surface potential measurements of the $NiO_x$ and $NiO_x$/SAM films by KPFM.

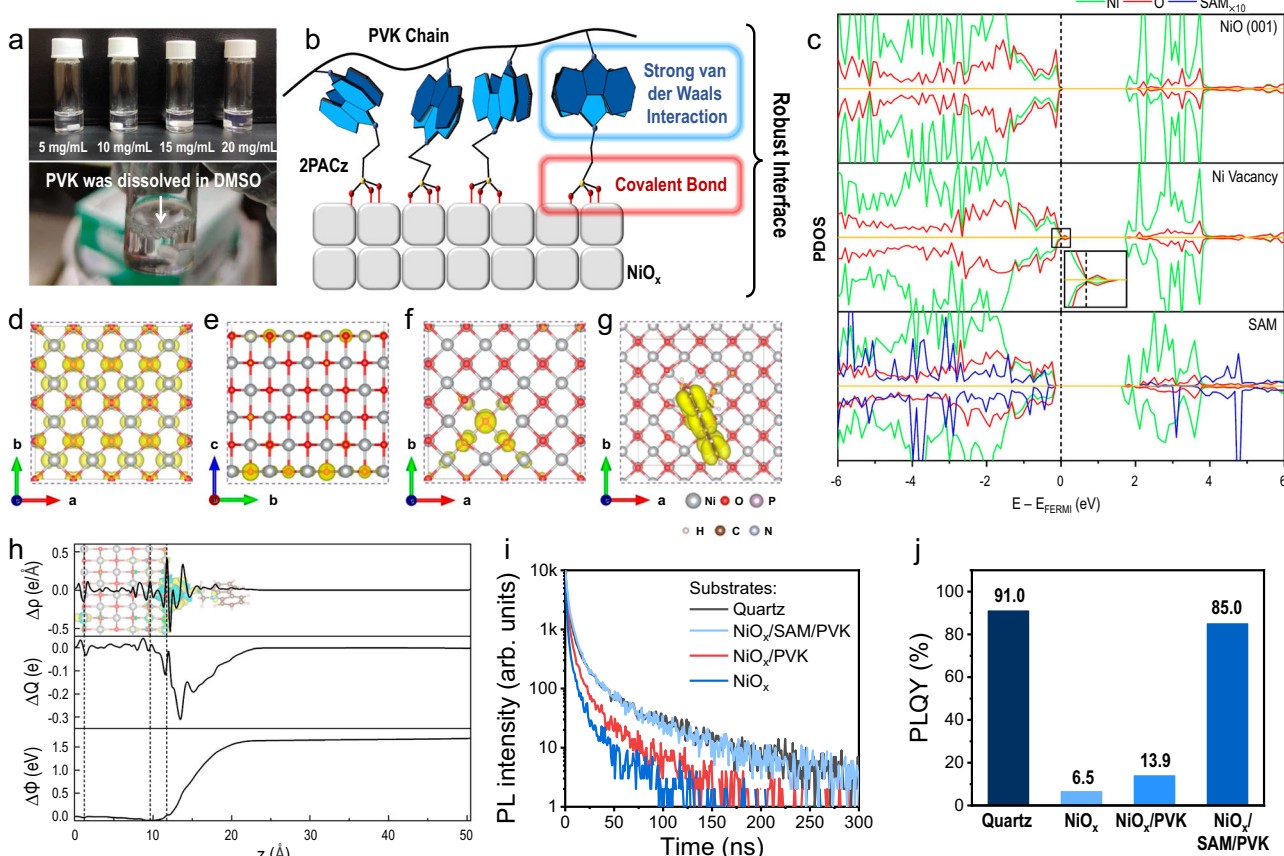

**Fig. 2 | Impact of PVK adhesion. a** Photos showing that PVK was dissolved in DMSO in a wide range of concentrations. **b** The schematic diagram showing the detailed configuration of NiO$_x$/SAM/PVK resulted in a robust interface. **c** Calculated PDOS for the pristine NiO$_x$ surface (top plot), the unpassivated NiO$_x$ surface with a Ni vacancy (middle plot) and the passivated NiO$_x$ surface with the SAM (bottom plot). The black dashed line indicates the location of the Fermi level, and '×10' indicates that the PDOS of the SAM molecules shown above is 10 times larger than the original PDOS to provide a clear demonstration. **d**, **e** The VBM isosurface plots of the pristine NiO$_x$ surface. **f** The unpassivated NiO$_x$ surface with a Ni vacancy. **g** The passivated surface with the SAM. **h** Calculated charge density difference ($\Delta\rho$), amount of charge transfer ($\Delta Q$) and electrostatic potential step ($\Delta\Phi$) at the NiO$_x$/SAM interface. PL lifetimes (**i**) and PLQYs (**j**) of (PEA/IPA)$_2$Cs$_{n-1}$Pb$_n$Br$_{3n+1}$ perovskites deposited on various substrates.

To study the adhesion property of the PVK layer to the NiO$_x$ and NiO$_x$/SAM substrates, we washed the PVK layer with DMSO and examined the changes in its morphology and thickness. As shown in Supplementary Figs. 4 and 5, the roughness of the PVK films increased after DMSO washing (from ~1.3 nm to ~3.3 nm in both cases), and the thickness of the PVK films was reduced from ~31 nm to ~5.4 nm and ~17 nm on the NiO$_x$ and NiO$_x$/SAM substrates, respectively. These results suggest that the DMSO had washed away a portion of the PVK film in both cases, but less severely for that on the 2PACz SAM substrate. We also assessed changes in the contact angle measurement. For the NiO$_x$/PVK film, after DMSO washing, the contact angle dramatically decreased from ~90° to ~56° (Supplementary Fig. 6), suggesting the exposure of the NiO$_x$ surface. However, the contact angle kept similar (~90°) in the NiO$_x$/SAM/PVK film, suggesting the remain of a compact PVK layer. To further examine the mechanical adhesion strength of the PVK/SAM and PVK/NiO$_x$ interfaces, we carried out a series of measurements as depicted in Supplementary Fig. 7. Our results indicate a greater force is required to detach the PVK layer from the SAM substrate as compared to the NiO$_x$ substrate. This finding suggests that the 2PACz layer significantly bolsters the adhesion strength of the PVK layer. Additionally, we observed that the average adhesion strength at the PVK/SAM interface is approximately 14% stronger than at the PVK/NiO$_x$ interface. We therefore conclude that the weakly interacted interface of NiO$_x$/PVK film is the origin of the susceptibility of PVK loss during the DMSO solvent processing. In

contrast, after introducing the SAM, the formation of covalent bonds between the NiO$_x$ surface and phosphonic acid groups in 2PACz, as well as the formation of stronger intermolecular van der Waals interactions (including $\pi$–$\pi$ stacking and dipole-dipole interaction, etc) between the 2PACz carbazole groups and the carbazole units in PVK (Fig. 2b), contribute to a more robust interface that alleviates the solvent washing effect to the PVK film[20–23].

To further investigate the impact of the dissolution of PVK film on perovskite emission, we deposited blue emissive quasi-2D (PEA/IPA)$_2$Cs$_{n-1}$Pb$_n$Br$_{3n+1}$ perovskite (PEA = phenylethylammonium, IPA = iso-propylammonium), which was well-developed in our previous work, onto the NiO$_x$, NiO$_x$/PVK and NiO$_x$/SAM/PVK substrates[5]. Supplementary Fig. 8 shows that increasing perovskite photoluminescence (PL) was observed in the order of NiO$_x$/SAM/PVK > NiO$_x$/PVK > NiO$_x$. The weakest PL in the NiO$_x$ case can be attributed to the strong PL quenching due to carrier trapping by the NiO$_x$ surface traps and fast hole transfer from perovskite to NiO$_x$. Adding a layer of PVK onto NiO$_x$ slightly suppressed these processes and led to stronger PL. However, in the NiO$_x$/SAM/PVK case, the PL dramatically enhanced, which can be attributed to the high coverage of the PVK film on the perovskite-coated NiO$_x$/SAM/PVK sample and the SAM-induced surface trap passivation of NiO$_x$ (which will be discussed later). Since the morphology and crystallinity of the perovskite films are comparable in the NiO$_x$/PVK and NiO$_x$/SAM/PVK cases (Supplementary Fig. 9a, b), we believe the change in film quality was not the key factor resulting in the

significantly improved PL. However, we still note that a slightly increased 3D perovskite ratio was observed in quasi-2D perovskite film formed on the $NiO_x$/SAM/PVK substrate (Supplementary Fig. 9c), resulting in a trivial redshift of PL emission (Supplementary Fig. 8).

To better understand the function of the SAM, we conducted first-principle calculations to study the role of the SAM in the $NiO_x$/SAM electronic structure (for computational details, see the "Methods" section). The projected density of states (PDOS) of both $NiO_x$ and the SAM-coated $NiO_x$ surface were obtained from density functional theory (DFT) calculations to investigate the trap passivation effect of the SAM, and the results are plotted in Fig. 2c. In agreement with previous studies on the pristine NiO surface[24], the valence band maximum (VBM) with a mixed Ni($d$)-O($p$) character was dominated by the delocalized surface states (Fig. 2d, e). A Ni vacancy on the surface led to the presence of shallow trap states above the VBM (Fig. 2c); the VBM wavefunctions were spatially localized at the under-coordinated oxygen atoms near the vacancy (Fig. 2f). The presence of 2PACz on the $NiO_x$ surface with a tridentate binding geometry successfully removed these shallow trap states (Fig. 2c). Our calculation results indicate that one of the three Ni atoms bound to 2PACz had a large displacement to fill the Ni vacancy, thus passivating the defect. This effect was also shown by the reduced sub-bandgap states of the $NiO_x$ surface after the introduction of SAM (Supplementary Fig. 10). In addition, as shown in Fig. 2h, due to the electron transfer from $NiO_x$ to 2PACz, the formed dipole at their interface led to an electrostatic potential step of +1.7 (−1.7) eV from $NiO_x$ to 2PACz for the electron (hole), which reduced the energy barrier for hole injection and affected charge transport across the SAM[25,26]. Within such a large energy barrier for electron injection, the electron tunneling possibility is relatively low[27–30]. Such a potential step played a role in blocking hole transfer directly from the perovskite towards $NiO_x$, which also contributed to suppressing PL quenching. Notably, this interfacial dipole also downshifted the $NiO_x$ Fermi level, leading to the alignment of the 2PACz HOMO and the $NiO_x$ VBM (Fig. 2c). As a result, the VBM of $NiO_x$/SAM showed hybridization between the wavefunctions of the $NiO_x$ surface and 2PACz (Fig. 2g). Additionally, the $p$-orbital energies of the O atoms adjacent to the vacancy were pushed from the VBM to deeper energy levels (Supplementary Fig. 11).

PL lifetime and PL quantum yield (PLQY) measurements were conducted to confirm the PL quenching effect further, as shown in Fig. 2i, j, respectively. The PL lifetimes of the perovskites were similar for quartz and $NiO_x$/SAM/PVK, reflecting that there was negligible PL quenching between perovskite and the hole transport layer. A decreased PL lifetime was found for the $NiO_x$/PVK sample that approached the lifetime of perovskite film on a bare $NiO_x$ substrate, suggesting that the thin PVK layer was insufficient to suppress PL quenching between perovskite and $NiO_x$. Similar results were observed for the PLQY measurements, whereby the highest value was obtained for the quartz (91%) sample and a slight drop to 85% for the $NiO_x$/SAM/ PVK sample. For the perovskite films coated on $NiO_x$/PVK and bare $NiO_x$, the PLQY significantly decreased to 13.9% and 6.5%, respectively, consistent with the PL lifetime results. The obvious quenching of perovskite emission would hinder the device performance of PeLEDs.

## Device performance of the PeLEDs

To investigate the impact of the organic/inorganic heterointerface on device performance, blue emissive PeLEDs with a device architecture of indium tin oxide (ITO)/$NiO_x$/(SAM)/PVK/perovskite/TPBi/LiF/Al were fabricated, and the energy levels are shown in Fig. 3a. With 2PACz modification, the VBM of the $NiO_x$ surface changed from −5.2 eV to −5.6 eV (Fig. 3b), approaching the HOMO of PVK (−5.7 eV). This energy level shift by ~0.4 eV agrees well with the work function difference measured by KPFM (Fig. 1c). It could be partially attributed to the formation of the aforementioned interfacial dipoles across the $NiO_x$/ SAM interface (Fig. 2h), while the carbazole moiety in the 2PACz SAM also contributed to the deepening of the energy level, thus reducing

the energy barrier of hole injection from $NiO_x$. Furthermore, the wavefunction of the resulting hybridized interface state was mainly delocalized at the 2PACz carbazole group and the $NiO_x$ surface, which effectively aided electronic coupling between the carbazole moieties of 2PACz and PVK, resulting in a cascade-type hole injection structure and smoothing the pathway for hole injection from $NiO_x$ to PVK.

These combined features contributed to a larger hole current improvement (Fig. 3c) and a dramatically decreased $V_T$ from 3.8 V (without SAM) to 2.1 V (with SAM) in the PeLED devices (Fig. 3d). Meanwhile, the improved hole injection increased the overall current density of the device and led to an ultra-high luminance of 10,392 cd m$^{-2}$ for the blue PeLEDs. Moreover, as shown in Fig. 3e, f, with SAM modification, the maximum/average EQEs of the PeLEDs improved from 8.9%/7.3% to 14.5%/13.0%, respectively, along with an improvement of the maximum current efficiency from 14.2 cd A$^{-1}$ to 23.2 cd A$^{-1}$ (Supplementary Fig. 13a), demonstrating one of the best EQEs with relatively high luminance (Table 1) among the published results (Fig. 3j). The findings indicate that the large improvement of the blue PeLEDs with SAM modification resulted from the following three effects: (i) suppressed emission quenching; (ii) smooth hole injection that effectively balanced the injected holes and electrons in the p-i-n device; and (iii) passivation of the $NiO_x$ surface traps by 2PACz, which minimized the loss of injected carriers. It is interesting to find that all the devices started to break down at a current density on a scale of ~10$^2$ mA cm$^{-2}$. This corresponded to breakdown voltages of 6.6 V, 7.6 V, 7.2 V, and 8.8 V for blue devices with and without SAM, and green devices (which will be discussed later) with and without SAM, respectively. We attribute this phenomenon to the generation of Joule heat under such high current densities, which destroyed the perovskite layers and caused the devices to break down. We also consider that the identity of the carbazole unit in both SAM and PVK might be critical for device performance improvement as when we changed 2PACz to MeO-2PACz that involves two more −OCH$_3$ groups in the carbazole unit, a poorer device performance was achieved (Supplementary Fig. 14).

In addition to benefiting the PeLED performance, the SAM introduction also facilitated more stable devices. As shown in Supplementary Fig. 15, the devices with and without the SAM both possessed excellent spectrum stability regardless of the applied voltage and while under continuous operation, which suggests that no change in the perovskite phase or stoichiometry occurred during the stability measurement. The half-lifetime of the device increases from 300 s to 831 s after SAM modification (Supplementary Fig. 16), which is at a moderate level compared with other published results (Supplementary Table 1). This result indicates that the robust organic/inorganic heterointerface contributed to the much-improved device stability. We preliminarily consider that this phenomenon might be related to the deformation of the PVK film under Joule heat during device operation owing to surface energy mismatch between PVK and $NiO_x$[31]. However, this hypothesis requires further in-depth investigation.

To evaluate the effect of the SAM on devices with other emission colors, we fabricated green-emissive PeLEDs [PEA$_2$(Cs$_{0.933}$FA$_{0.067}$)$_{n-1}$ Pb$_n$Br$_{3n+1}$] based on the same device architecture. As shown in Table 1 and Fig. 3g−i, similar to their blue counterparts, a significant improvement in device performance was observed. The maximum/ average EQE increased from 21.6%/19.3% (without SAM) to 26.0%/ 23.8% (with SAM), leading to an improvement of maximum current efficiency from 69.6 cd A$^{-1}$ to 91.0 cd A$^{-1}$ (Supplementary Fig. 13b) and a luminance improvement from 45,100 cd m$^{-2}$ to 83,561 cd m$^{-2}$. This high device performance makes our green PeLED one of the state-of-the-art devices in this field (Fig. 3j). Moreover, all the green PeLEDs possessed good spectrum stability; their emission maintained at 515 nm under various applied voltages (Supplementary Fig. 17). Additionally, $V_T$ was reduced to 2.1 V in both the blue and green devices, suggesting that the energy level coupling at the $NiO_x$/SAM/PVK interface was already close to ideal, and the holes could efficiently inject

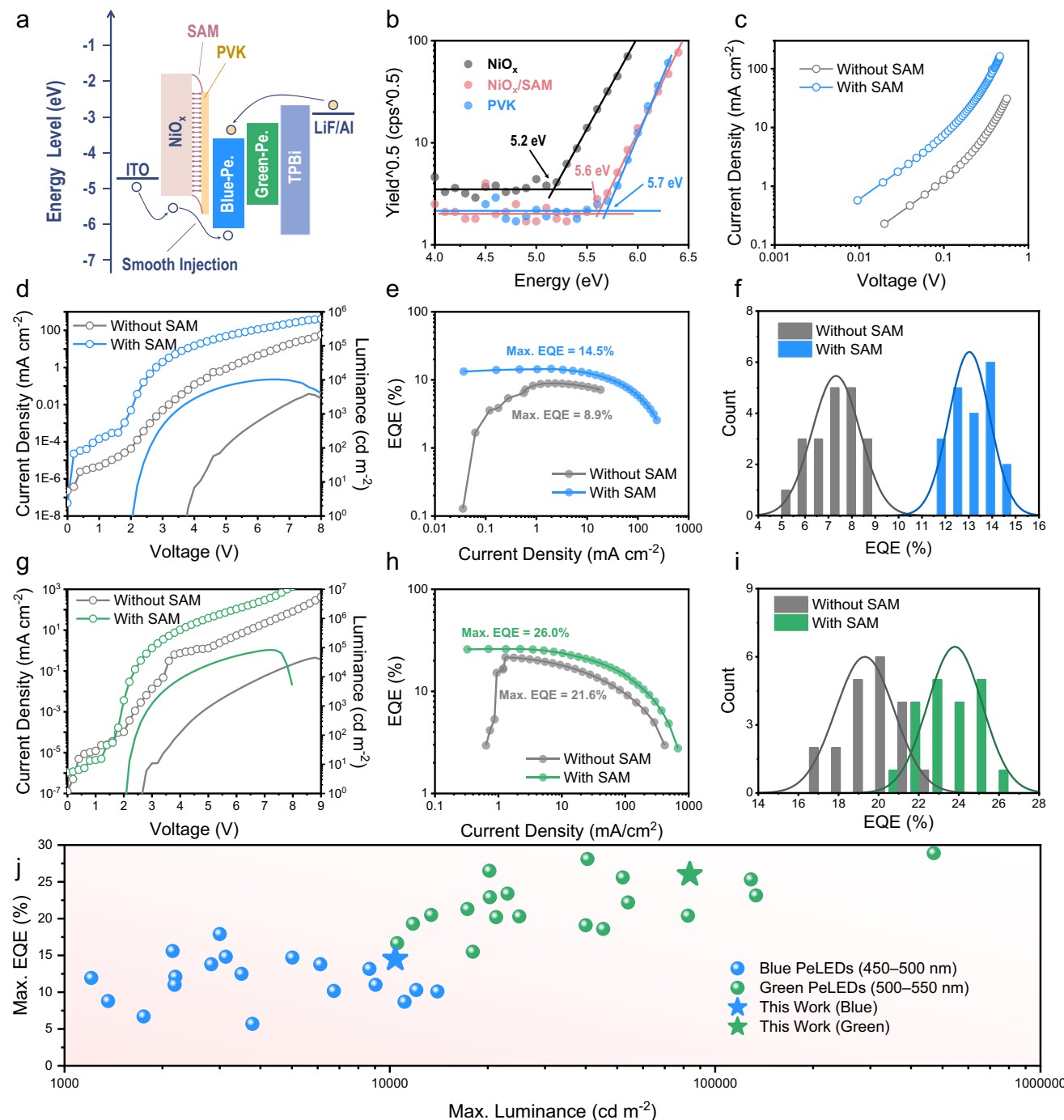

**Fig. 3 | Device performance of blue (EL peak at 493 nm) and green (EL peak at 515 nm) PeLEDs. a** Energy level diagram for different component layers in PeLED. The energy levels of blue and green perovskites were acquired from our previous work (ref. 5) and Supplementary Fig. 12, respectively. **b** Atmospheric ultraviolet photoelectron spectroscopy spectra indicating VBM/HOMO levels of NiO_x, NiO_x/SAM and PVK. **c** Hole-only devices with the structures of ITO/NiO_x/(SAM)/PVK (DMSO washed)/MoO_3/Ag. Current density–voltage curves (dotted lines) and luminance–voltage curves (solid lines) of the best (**d**) blue and (**g**) green PeLEDs with and without SAM. Current density–EQE curves of the best (**e**) blue and (**h**) green PeLEDs with and without SAM. Statistical EQEs of 20 (**f**) blue and (**i**) green PeLEDs with and without SAM. **j** Summary of selected device performances of blue (EQE > 5% and luminance >1000 cd m$^{-2}$) and green (EQE > 15% and luminance >10,000 cd m$^{-2}$) PeLEDs published in the past 4 years (2019–2023). Data are from refs. 3, 4, 51–86.

through the interface. Our results indicate that modification of the organic/inorganic heterointerface is a universal solution that can boost the efficiency of blue and green PeLEDs.

### Device response of the PeLEDs

PeLEDs are reported as light signal generators in optical communication by defining the light 'on' and 'off' states as '1' and '0',

respectively[32,33]. Considering the high mobility and bipolar nature of perovskites, as well as their nanosecond-scale radiative decay, PeLEDs have the potential to generate light signals at a high speed of up to >1 gigabit per second; therefore, they are promising for use in optical communication techniques such as light fidelity[34]. In addition to the radiative lifetime of a perovskite emitter, the charge injection and transport properties of PeLEDs are other important factors governing

the response speed of the device, which becomes the main limitation at the current development stage and can result in a relatively low cut-off frequency (i.e., a frequency of −3 dB) in the kHz scale[35].

To investigate the role of the 2PACz SAM in device response, we applied a square-wave voltage of 7 V with a 50% duty ratio to the green PeLEDs (which provided a much higher luminance than their blue counterparts), and guided the EL emission to a silicon photodetector connected to an oscilloscope. Figure 4a shows that the overall response times of the devices with and without SAM were 373 μs and 540 μs, respectively, demonstrating that the introduction of the SAM accelerated the device response, which is further confirmed by the statistic results (Supplementary Fig. 18). Furthermore, we varied the frequency of the square-wave voltage from 100 Hz to 10,000 Hz, and the results are presented in Fig. 4b and Supplementary Fig. 19. The cut-off frequency of the device with SAM (2940 Hz) was more than two times higher than that of the device without SAM (1322 Hz), consistent with the faster response speed results.

Electrochemical impedance spectroscopy (EIS) study was performed to characterize the charging and discharging properties of

carriers in PeLEDs as the rise and fall time is strongly related to the device resistance and capacitance (i.e., the resistor-capacitor step response). The results are presented as Nyquist plots in Fig. 4c. According to the circuit model used for EIS analysis (Fig. 4d)[14], the series resistance ($R_s$), transport resistance ($R_t$), recombination resistance ($R_r$), transport capacitance ($C_t$) and recombination capacitance ($C_r$) were reduced from 13.9 Ω, 35.6 Ω, 4340 Ω, 12 nF and 4.5 nF to 7.1 Ω, 8.3 Ω, 172 Ω, 2.0 nF, 4.1 nF, respectively, after introducing SAM. The significantly reduced $R_t$ and $C_t$ can be attributed to the smoother charge injection property of the NiO$_x$/SAM/PVK-based PeLEDs (discussed above), which decreased the time constant ($R_t \times C_t$) and hence facilitated a faster-responding device. The reduced $R_r$ and $C_r$ can be attributed to the passivation of NiO$_x$ surface traps by 2PACz. Consequently, we conclude that charge injection and transport are the main factors limiting the response speed in our PeLEDs, and the introduction of SAM, which successfully reduces transport resistance and capacitance, is an effective method to increase the device response.

## Discussion

In this work, we unexpectedly found that the commonly used hole injection NiO$_x$/PVK bilayer films suffered from a processing issue, whereby the solvent of the perovskite precursor solution (DMSO) washed away the PVK owing to weak physical interaction at this organic/inorganic heterointerface. The partially removed PVK molecules led to NiO$_x$ surface exposure, which quenched the emission from the upper perovskite layer. By introducing a 2PACz SAM to form a NiO$_x$/SAM/PVK hole injection structure, we successfully alleviated PL quenching owing to the following three aspects: (i) the formation of a robust interface via stronger van der Waals forces between the SAM and PVK, which improved PVK adhesion; (ii) passivation of the NiO$_x$ surface traps by 2PACz molecules, which suppressed non-radiative

**Table 1 | Summary of device performances of blue and green PeLEDs with and without SAM**

| PeLED | With/ without SAM | $V_T$ (V) | Max luminance (cd m$^{-2}$) | Max EQE (%) | Max current efficiency (cd A$^{-1}$) |
|---|---|---|---|---|---|
| Blue (EL = 493 nm) | w/o | 3.8 | 2105 | 8.9 | 14.2 |
| | w/ | 2.1 | 10,392 | 14.5 | 23.2 |
| Green (EL = 515 nm) | w/o | 2.7 | 45,100 | 21.6 | 69.6 |
| | w/ | 2.1 | 83,561 | 26.0 | 91.0 |

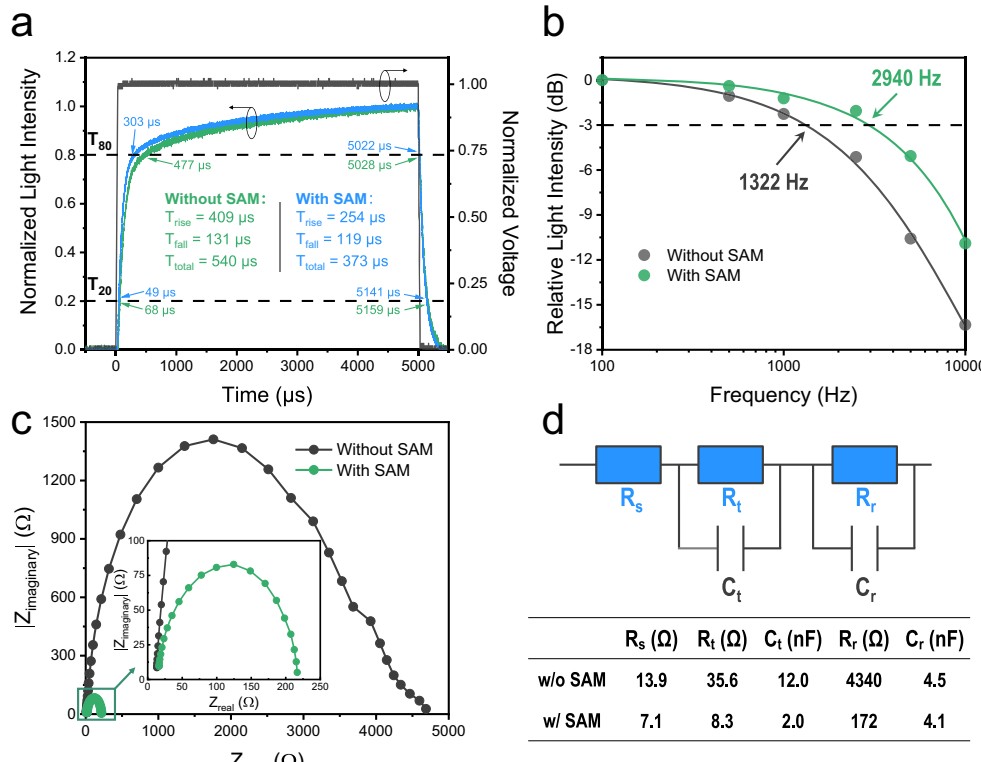

**Fig. 4 | Device response properties of green PeLEDs with and without SAM. a** Response time ($T_{total} = T_{rise} + T_{fall}$) of different devices at 100 Hz, where $T_{rise} = T_{80} − T_{20}$ is the rise time of light intensity after the voltage is on, and $T_{fall} = T_{20} − T_{80}$ is the fall time of light intensity after the voltage is off. **b** Relative light intensity–frequency curves of different devices. The intersections with the −3 dB line represent the cut-off frequencies. **c** EIS measurement of different devices. The inset is a zoom-in figure of the marked area. **d** Circuit model used for the EIS analysis, and the extracted values of each component.

recombination; and iii) formation of an interfacial dipole, which blocked the back transfer of holes from the perovskite to $NiO_x$.

The reduced $V_T$ and increased current density of the PeLEDs constructed with the 2PACz SAM were attributed to a cascade-type structure for more efficient hole injection from $NiO_x$ to PVK, improving the device performance. The best blue and green PeLEDs achieved an EQE and luminance of 14.5% and 10,392 cd m$^{-2}$, as well as 26.0% and 83,561 cd m$^{-2}$, respectively, representing some of the best-performing blue and green PeLEDs. In addition, the PeLEDs constructed with a SAM had much faster response speeds and more than two times higher cut-off frequencies than those without a SAM, which suggests that interfacial engineering is an effective method to increase the device response. This work demonstrates a simple but effective strategy to improve the organic/inorganic hole injection heterointerface for more efficient and brighter PeLEDs with quick response, paving the way for new applications of PeLEDs, such as light fidelity.

## Methods

### Materials and chemicals

Nickel(II) acetate tetrahydrate [Ni(CH$_3$COO)$_2$·4H$_2$O, 99.995%], ethanolamine (NH$_2$CH$_2$CH$_2$OH, 99.5%), ethanol (99.8%), poly(9-vinylcarbazole) (PVK, average M$_n$ ranging from 25,000 to 50,000 g mol$^{-1}$), CsBr (99.999%), PbBr$_2$ (99.999%), chlorobenzene (anhydrous, 99.8%), and ethyl acetate (anhydrous, 99.8%) were purchased from Sigma-Aldrich. 2PACz (>98.0%) was purchased from Tokyo Chemical Industry Co., Ltd. Phenylethylammonium bromide (PEABr, 99.5%), *iso*-propylammonium bromide (IPABr, 99.5%) and formamidinium bromide (FABr) were purchased from Xi'an Polymer Light Technology Corp. DMSO (>99.0%) was purchased from Acros. LiF was purchased from Alfa Aesar. 1,3,5-Tris(1-phenyl-1H-benzimidazol-2-yl)benzene (TPBi) was purchased from Lumtec. All chemicals were used as received.

### Solution preparation

Blue perovskite precursor solution was prepared using our previous formula[5]; briefly, CsBr (19.2 mg), PbBr$_2$ (55.1 mg), PEABr (24.2 mg), and IPABr (8.4 mg) were dissolved in DMSO (1 mL). For the green perovskite precursor solution, CsBr (89.4 mg), FABr (3.8 mg), PbBr$_2$ (110.1 mg) and PEABr (24.2 mg) were dissolved in DMSO (1 mL). For the PVK solution, 8 mg of PVK was dissolved in 1 mL of chlorobenzene. For the 2PACz solution, 1 mg of 2PACz was dissolved in 4 mg of ethanol. All of the above solutions were stirred for 12 h at room temperature. For the $NiO_x$ precursor solution, a 1:1 molar ratio of Ni(CH$_3$COO)$_2$·4H$_2$O and NH$_2$CH$_2$CH$_2$OH was dissolved in ethanol under continuous stirring for 12 h at 70 °C, keeping the concentration of Ni$^{2+}$ at 0.1 M.

### PeLED fabrication

Patterned ITO-coated glass substrates (15 mm × 15 mm) were cleaned by sequential sonication in detergent, deionized water, acetone and isopropyl alcohol and then dried at 65 °C in a baking oven. After a 5-min oxygen plasma treatment of ITO substrates, the $NiO_x$ layers were deposited onto these ITO substrates by spin-casting the 0.45 μm PTFE-filtered $NiO_x$ precursor solution at 3000 rpm for 50 s and then baking them at 270 °C for 45 min in ambient air. After cooling to room temperature, the $NiO_x$-coated substrates were subjected to another 5-min oxygen plasma treatment to modify their surface properties (e.g., increasing the surface wettability, electrical conductivity, and work function)[36,37]. The PVK solution was then spin-coated (2000 rpm for 30 s) onto the substrates, and the resulting coated substrates were annealed at 150 °C for 30 min in an N$_2$-filled glovebox. For the device containing the 2PACz SAM, before depositing the PVK layer, the 2PACz solution was spin-coated (2000 rpm for 30 s) onto the $NiO_x$ substrate, followed by annealing at 100 °C for 10 min. For the blue perovskite emitter, the corresponding precursor solution was then spin-coated onto the PVK layer at 4000 rpm for 2 min, and ethyl acetate (100 μL)

was introduced as an anti-solvent at 26 s after the beginning of spin-coating, followed by annealing at 70 °C for 10 min. For the green perovskite emitter, the corresponding precursor solution was spin-coated onto the PVK layer at 4000 rpm for 90 s, followed by annealing at 110 °C for 10 min. Finally, TPBi (32 nm) and LiF/Al electrodes (1 nm/100 nm) were deposited using a thermal evaporation system under a high vacuum of <1 × 10$^{-6}$ Torr. LiF/Al electrode is selected because the chemical reaction between LiF and Al (forming chemical bonds of Al–F at the interface), can reduce the work function of this electrode to ~2.9 eV[38]. The device's active area was 8 mm$^2$, as defined by the overlapping area of the ITO and Al electrodes.

### Film and device characterization

XPS data were recorded on a Thermo Scientific ESCALAB 250Xi spectrometer under 2 × 10$^{-10}$ Pa and X-ray source (XR6 monochromated Al Kα, hν = 1486.68 eV) was exerted with a 500 μm spot size and 20 eV pass energy. Steady-state PL was recorded using a Horiba Fluorolog system (Horiba Ltd., Japan) equipped with a single grating, using a monochromatized Xe lamp as the excitation source. Transient PL decay was measured using a Quantaurus-Tau fluorescence lifetime measurement system with an excitation wavelength of 365 nm (C11367-03, Hamamatsu Photonics Co., Japan). The PLQYs of the perovskite films were recorded on a commercial PLQY measurement system (Ocean Optics) with excitation from a 365-nm LED. The contact angle tests were performed on a DataPhysics OCA40 microsurface contact angle analyzer. The tensile test was based on a tensile machine developed by Dongriyiqi (DR-509AQ). The samples were prepared with a structure of Glass (15×15 mm)/ITO/$NiO_x$/SAM/PVK. Then one end of a 3 M scotch tape (width of 12.7 mm) was firmly stuck on the surface of PVK. The other end of the tape was fixed on the up holder of the tensile machine for pulling up. Then a double-side tape was stuck to the other side of the sample and firmly mounted on the bottom holder. During pulling up of the up holder, the force-displacement relationship was detected by highly sensitive force and spatial sensors, respectively, and recorded by a computer. Atmospheric ultraviolet photoelectron spectra were recorded on an AC-3 system with a standard deviation of ±0.02 eV (Riken Keiki Co., Ltd). Surface potential measurement was carried out using KPFM (Bruker Dimension Icon). In our experiment, both samples were measured by the same KPFM reference tip with a work function of 4.42 eV. Along with the surface potential ($V_{CPD}$), the work function of the samples ($\varphi_{sample}$) can be calculated by equation of $V_{CPD} = \frac{\varphi_{tip} - \varphi_{sample}}{e}$. The crystalline structures of the perovskite films were investigated using an X-ray diffractometer (PANalytical X'pert PRO, Netherlands) equipped with a Cu-Kα X-ray tube. The SEM images were obtained with a Regulus 8100 SEM (Hitachi, Japan). The AFM measurements were carried out using a Digital Instrumental Multimode Nanoscope IIIa in tapping mode. The absorption spectra were recorded on a UV-Vis spectrometer (Ocean Optics QE65 Pro). Photothermal deflection spectroscopy measurements were carried out with a 1 kW Xe arc lamp and a 1/4 m grating monochromator (Oriel) as the tunable light source. The pump beam was modulated at 13 Hz by a mechanical chopper before irradiating the sample. Perfluorohexane was used as the deflection fluid. A Uniphase HeNe laser as the probe beam was directed parallel to the sample surface. A quadrant cell (United Detector Technology) was used as the position sensor for monitoring the photothermal deflection signal of the probe beam. The output signal of the detector was fed into a lock-in amplifier (Stanford Research, Model SR830) for phase-sensitive measurements. All photothermal deflection spectroscopy spectra were normalized to the incident power of the pump beam.

The EIS measurements were carried out using a Paios 4.0 Measurement Instrument (FLUXiM AG, Switzerland). The current density–voltage and luminance–voltage curves, EL spectra, and EQE of the PeLED were recorded simultaneously on a commercial system (XPQY-EQE-350-1100, Guangzhou Xi Pu Optoelectronics Technology

Co., Ltd., China) that was equipped with an integrating sphere (GPS-4P-SL, Labsphere) and a photodetector array (S7031-1006, Hamamatsu Photonics). Device operating lifetime was also measured by this system. The absolute spectral radiant flux is calibrated by using a NIST-traceable radiant-flux standard lamp (HL-3P-INT-CAL, Ocean Optics Co., Ltd.). The same measurement and calibration can also be found in other literature using the same instrument[37]. The stability test was carried out in a $N_2$-filled glovebox without encapsulation (25 °C, $H_2O$ concentration lower than 0.1 ppm). The setup to record the responses of devices was composed of a signal generator (DG 4202, Rigol), a silicon photodetector (DET10A2, Thorlabs), and an oscilloscope (DS7024, Rigol). All of the device characterization tests for the PeLEDs were carried out in an $N_2$-filled glovebox.

## DFT calculations

Owing to the antiferromagnetic nature of NiO-based structures, the spin-polarized first-principle calculations based on DFT were performed using the Vienna ab initio simulation package (VASP)[39,40]. The projector-augmented wave pseudopotentials[41,42] were applied to treat the valence and core electron interaction, and a kinetic energy cut-off of 500 eV was adopted for the plane wave basis set. The electron correlation and exchange effects were described by the Predew–Burke–Ernzerhof functional under the generalized gradient approximation (GGA) scheme[43]. The strongly localized electrons in the d-shells of Ni atoms were simulated using the GGA + U method[44]. According to Cococcioni and de Gironcoli's work for NiO, we set an effective Hubbard parameter $U_{eff}$ of 4.60 eV on the basis of the linear response method in our calculations[45]. To investigate the trap passivation and functionalization effects of the SAM molecules on the NiO surface, we built a surface slab for NiO (001) consisting of six layers in a (4 × 4) supercell. The model with one Ni vacancy and a tridentate binding configuration of the SAM molecules on the surface was adopted[46]. All of the slabs were periodically repeated in the three dimensions. To minimize the interaction between neighboring slabs, a vacuum space of at least 25 Å was added on the top of the slab surface. A compensating dipole was introduced to cancel spurious dipolar coupling between the adjacent replicas along the direction perpendicular to the slab[47]. The vdW interactions were taken into account based on Grimme's D3 correction[48]. The Brillouin zone was sampled with a 2 × 2 × 1 grid of k-points centered at gamma. The energy and force convergence criteria were set to $10^{-5}$ eV and 0.01 eV Å$^{-1}$, respectively. Data post-processing was conducted using the VASPKIT and JAMIP packages[49,50].

## Data availability

The data that support the findings of this study are provided in the main text and the Supplementary Information. The original data are available from the corresponding authors upon request.

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

## Acknowledgements

This study was financially supported by the Hong Kong Research Grant Council for the GRF grant (No. 11314122), the National Natural Science Foundation of China (No. 62075065) and the Guangdong Major Project of Basic and Applied Basic Research (No. 2019B030302007). Z.C. is a Marie Skłodowska-Curie Postdoctoral Fellow (Project No.: 101064229) funded by UK Research and Innovation (Grant Ref.: EP/X027465/1). X.-K.C. acknowledges the New Faculty Start-up Grant of the City University of Hong Kong (7200709, 9610547). X.-K.C. also acknowledges the supports from Suzhou Key Laboratory of Functional Nano & Soft Materials, Collaborative Innovation Center of Suzhou Nano Science & Technology, and the 111 Project.

## Author contributions

Z.C. and H.-L.Y. conceived the idea for the study and designed the experiments. Z.L., L.C., and Z.C. fabricated the blue and green PeLEDs and carried out the device characterizations. Z.L., Z.C., G.Z., and L.C. carried out the perovskite film characterizations. Z.S. carried out the DFT calculations under the supervision of X.-K.C.; C.Z. carried out the photothermal deflection spectroscopy measurements under the supervision of S.S. Z.C., Z.S., Z.L., X.-K.C., and H.-L. Y. analyzed the data and wrote the manuscript. Z.C. and H.-L.Y. led the project. All authors contributed to the manuscript.

## Competing interests

The authors declare no competing interests.

## Additional information

[1]State Key Laboratory of Luminescent Materials and Devices, Institute of Polymer Optoelectronic Materials and Devices, School of Materials Science and Engineering, South China University of Technology, 381 Wushan Road, Guangzhou 510640, P. R. China. [2]State Key Laboratory of Advanced Materials and Electronic Components, Guangdong Fenghua Advanced Technology Holding Co. Ltd., Zhaoqing, Guangdong 526020, China. [3]Department of Chemistry and Centre for Processible Electronics, Imperial College London, London W12 0BZ, UK. [4]Department of Chemistry, City University of Hong Kong, Tat Chee Avenue,

Kowloon, Hong Kong. [5]Department of Materials Science and Engineering, City University of Hong Kong, Tat Chee Avenue, Kowloon, Hong Kong. [6] Hong Kong Institute for Advanced Study, City University of Hong Kong, Tat Chee Avenue, Kowloon, Hong Kong. [7]Institute of Functional Nano & Soft Materials (FUNSOM), Soochow University, Suzhou 215123 Jiangsu, PR China. [8]Jiangsu Key Laboratory of Advanced Negative Carbon Technologies, Soochow University, Suzhou 215123 Jiangsu, PR China. [9]Department of Physics and Institute of Advanced Materials, Hong Kong Baptist University, Kowloon Tong 999077 Hong Kong SAR, P.R. China. [10]School of Energy and Environment, City University of Hong Kong, Tat Chee Avenue, Kowloon, Hong Kong. [11]Hong Kong Institute for Clean Energy, City University of Hong Kong, Tat Chee Avenue, Kowloon, Hong Kong. [12]These authors contributed equally: Zhenchao Li, Ziming Chen.
✉e-mail: z.chen@imperial.ac.uk; xkchen@suda.edu.cn; a.yip@cityu.edu.hk

