## [Peer Review File · Nature Communications]

Charge Injection Engineering at Organic/Inorganic
Heterointerfaces for High-Efficiency and Fast-Response
Perovskite Light-Emitting DiodesREVIEWER COMMENTS

Reviewer #1 (Remarks to the Author):

Metal-halide perovskites are promising semiconducting materials for optoelectronic applications due to their excellent optical and electronic properties. Quasi-two-dimensional perovskites, in particular, offer a feasible way to modulate their bandgaps and electronic structures, leading to high external quantum efficiencies for PeLEDs. However, the inorganic/organic heterointerfaces in PeLEDs often result in reduced physical robustness and imperfect hole injection. To overcome this, a self-assembled monolayer of [2-(9H-carbazol-9-yl)ethyl]phosphonic acid was introduced as a bridge to form a NiO_x/2PACz/PVK tri-layer hole transport/injection layer, resulting in improved performance for blue and green PeLEDs, including reduced turn-on voltages, higher EQEs, and faster response speeds. This work can be published but the following comments should be carefully addressed with a major revision, before being considered for publication.

1) SAM characterization. Contact angle does not give detailed information about the bonding nature. Photoelectron spectroscopic analysis is required to quantitatively and qualitatively confirm the formation of the desired interface as illustrated in figure 1a. The surface coverage can be also determined by the photoelectron spectroscopy.

2) The intermolecular interaction between PVK and carbazole monolayer was defined as van der Waals. What about pi-pi staking or dipole-dipole interaction? To me, these are better suitable than van der Waals. Describe more details about how it can be defined as van der Waals or provide the relevant evidence.

3) A big concern to me is lack of discussion on charge transport across SAM although the central theme of this work is using SAM for hole injection layer. Charge transport across SAM can depend on the dipole of SAM (refs: J. Phys. Chem. Lett. 2018, 9, 17, 5078–5085; J. Am. Chem. Soc. 2019, 141, 22, 8969–8980;). However, it has been also reported that the dipole of individual molecule can be canceled out in SAM (mostly done by Whitesides group; refs: J. Am. Chem. Soc. 2014, 136, 1, 16–19; Angew. Chem. Int. Ed. 2012, 51, 4658-4661; J. Am. Chem. Soc. 2015, 137, 11, 3852–3858; JACS, 2012, 134, 10876–10884). These discussions on the role of dipole on charge transport across SAM were expected but lacked. These thus must be added with the extensive reference citations.

4) The following studies involving SAM in perovskite interface engineering are directly suitable to this work and thus should be cited: Adv. Opt. Mater. 2022, 10, 1, 2101361; Adv. Energy Mater. 2020, 10, 44, 2002606; Nano Energy, 2020, 73, 104752.

5) Comparison of surface potential using KPFM is often misleading as the reference is missing.

6) Figure 2b means a lot. First, it means carbazole terminal groups are loosely packed (i.e., SAM is disordered largely) otherwise the counterpart carbazole substituent from PVK polymer cannot be intercalated as illustrated. Second, the loosely packed SAM is free from the dipole cancellation maximizing the dipole effect on WF. However, I cannot find any evidence or justification that supports the Figure 2b. Is this because of the surface of NiOx substrate is rough causing the disorder of SAM? Or, the SAM is prepared by spin-coating for a short time (30 seconds), which can give sparse SAM rather than dense one, which is not sure intended or not.

Reviewer #2 (Remarks to the Author):

In this manuscript, Zhenchao Li et al. introduced a SAM layer between NiOX and PVK to build a robust interface, passivate interfacial deep traps, and align the energy levels. Their as-fabricated blue and green PeLEDs with external quantum efficiencies of 14.5% and 26.0%, respectively. The SAM layer also enhances the devices' response speed. Overall, this work is impressive and interesting. I found some major issues in these tests and analyses, as described below.

1. There are a lot of available SAM layers, such as 2PACz, 3PACz, 4PACz. Why are the authors focused on 2PACz? The comparison among various SAM layers are recommended to enrich this manuscript.
2. Why choose the Al electrode for perovskite devices? Will Al ions migrate through the device and react with perovskites?
3. "After cooling to room temperature, the NiOx-coated substrates were subjected to a 5-min oxygen plasma treatment". Why is the oxygen plasma treatment needed here?
4. The photos for the blue and green PeLED devices are needed in the supplementary information.
5. For Supplementary Figure 5, what caused the difference in the positions of the PL emission peaks?
6. For the device stability test (Supplementary Figure 12), what are the test conditions (temperature, humidity...)? This important information should be disclosed in the Methods section so that other groups can reproduce the authors' results.
7. Although SAM enhances the devices' lifetime, it is still relatively short (<1 h). Have the authors done any encapsulation for the as-fabricated PeLEDs? The short lifetime prevents PeLED from being a real application. Are there any other factors limiting devices' lifetime?
8. A summary table on the lifetime of reported PeLED is suggested. We should pay attention to devices' stability just like perovskite solar cell field.

9. The SAM also improves the response speed of PeLED. However, statistical data is needed to reach a reliable conclusion.
10. Is it possible to measure the mechanical adhesion of the device with and without a SAM layer?
11. What are the breaking down voltages for blue and green PeLEDs?
12. How do the authors measure the EQE and luminescence of PeLEDs? The calibration step is important for the accuracy of results.

Based on these main concerns, I think the quality of this manuscript is not good enough so far. I recommend further consideration for publication of this manuscript in Nature Communications after major revision.

Reviewer #3 (Remarks to the Author):

This manuscript reports on an approach to fabricate high-efficiency and fast-response perovskite LED by inserting a self-assembled monolayer (SAM) of [2-(9H-carbazol-9-yl)ethyl]phosphonic acid (2PACz) between NiOx and poly(9-vinylcarbazole) (PVK) layers. SAM improves robustness between NiOx and PVK layers, tunes energy level and passivates surface traps of NiOx layer. Perovskite LEDs using the tri-layer of NiOx/SAM/PVK have high EQE and luminance for both blue and green emissions, and also have significantly decreased turn-on voltage to 2.1 V and fast response time. Overall this is an interesting work but there are several important points that need to be addressed first. Detailed comments are as follows.

1. In Fig. 1c, surface potentials of NiOx and NiOx/SAM layers from KPFM look similar and uniform in the current scale. Narrowing the scale bar would be helpful for comparing the surface potential and morphology more clear.
2. In Fig. 2b, PVK has strong interaction with 2PACz, so PVK can remain more than without 2PACz after the DMSO washing. However, there is no direct evidence for the strong interaction between PVK and 2PACz layers. So, adding the crystallinity or electrical property would be nice to support the robustness or strong interaction between 2PACz and PVK layers.
3. To claim “better adhesion” between the SAM and PVK layer, mechanical peeling experiment should be performed.
4. In supplementary fig. 5, PL emission wavelength is red shifted from NiOx to NiOx/PVK and NiOx/SAM/PVK. Even if the authors explain that there is a negligible impact on perovskite crystallinity using XRD, this red shift can be explained by more 3D phase in perovskite film on NiOx/SAM/PVK substrate. The explanation about this wavelength shift would be helpful for the impact on the perovskite layer by different substrates.

5. In the introduction, the new record red LED based on quasi-2D perovskites with over 26% EQE should be mentioned (Nature Communications 2023, 14, 397).

6. In the abstract and introduction, the authors should state the wavelength (or clarify it is sky-blue rather than pure blue) of the blue LED before claiming the efficiency of 14.5%. In addition, EL spectra should be provided in the main figures to avoid confusion.

REVIEWER COMMENTS

Point-by-point response of reviewers' comments

Reviewer #1 (Remarks to the Author):

Metal-halide perovskites are promising semiconducting materials for optoelectronic applications due to their excellent optical and electronic properties. Quasi-two-dimensional perovskites, in particular, offer a feasible way to modulate their bandgaps and electronic structures, leading to high external quantum efficiencies for PeLEDs. However, the inorganic/organic heterointerfaces in PeLEDs often result in reduced physical robustness and imperfect hole injection. To overcome this, a self-assembled monolayer of [2-(9H-carbazol-9-yl)ethyl]phosphonic acid was introduced as a bridge to form a NiO_x/2PACz/PVK tri-layer hole transport/injection layer, resulting in improved performance for blue and green PeLEDs, including reduced turn-on voltages, higher EQEs, and faster response speeds. This work can be published but the following comments should be carefully addressed with a major revision, before being considered for publication.

Reply:

We thank the reviewer for their very positive comments. We have made a major revision to further improve the scientific quality of this manuscript, and believe the revised manuscript is suitable to be published in Nature Communications.

1) SAM characterization. Contact angle does not give detailed information about the bonding nature. Photoelectron spectroscopic analysis is required to quantitatively and qualitatively confirm the formation of the desired interface as illustrated in figure 1a. The surface coverage can be also determined by the photoelectron spectroscopy.

Reply:

We thank the reviewer for their comments.

We agree with the reviewer that photoelectron spectroscopy is a powerful tool to reveal the interaction between materials. Therefore, we conducted X-ray photoelectron spectroscopy (XPS) studies to better understand how 2PACz molecules interact with NiO_x and form the desired interface. We prepared two samples including ITO/NiO_x and ITO/NiO_x/2PACz for comparison. In

Supplementary Figures 1a and 1b, compared with the ITO/NiO_x sample, the P 2p signal at 133.2 eV and N 1s signal at 399.2 eV confirm the presence of 2PACz molecules on NiO_x surface. In Supplementary Figures 1c and 1d, the high-resolution spectrum of the O 1s region elucidates the contribution of three different oxygen species. The fitted peak 1 (~529.2 eV), peak 2 (~530.8 eV), and peak 3 (~532.2 eV) are attributed to a typical Ni–O bond in NiO crystal, low-oxygen-coordinated defect site and the surface-adsorbed oxygen species, as well as interfacial interaction between Ni and O (from organic groups), respectively (ACS Omega 2022, 7, 12147–12157; Dalton Trans. 2017, 46, 9201–9209). After the deposition of SAM, the relative intensity of peak 3 apparently increases, suggesting a large amount of 2PACz molecules interact with NiO_x surface via the Ni–O interactions. In Supplementary Figures 1e and 1f, the whole Ni 2p spectrum shifts towards a higher binding energy side by ~0.3 eV after the SAM treatment. The observed chemical shift suggests that the oxygen atoms in 2PACz are contributing lone pairs of electrons to the vacant orbitals of Ni atoms. As a result, coordinate covalent bonds of P=O···Ni are formed at the interface, as illustrated in Fig. 1a (Sol. RRL 2021, 5, 2100663). We therefore added the following discussion in the revised manuscript to explain this interaction clearly.

On page 3:

‘The phosphonic acid groups in 2PACz can form robust tridentate bonds with the NiO_x surface¹⁷, as illustrated in Fig. 1a, which is also confirmed by the X-ray photoelectron spectroscopy (XPS) studies (Supplementary Figure 1).’

In Supplementary Information:

Supplementary Figure 1 | XPS analysis of the NiO_x and NiO_x/SAM samples. **a** The P 2*p* signal. **b** The N 1*s* signal. **c** and **(d)** the O 1*s* signal of NiO_x and NiO_x/SAM samples, respectively. **e** The Ni 2*p* signal and **(f)** the Zoom-in Ni 2*p* signal of NiO_x and NiO_x/SAM samples, respectively. Compared with the ITO/NiO_x sample, the P 2*p* signal at 133.2 eV and N 1*s* signal at 399.2 eV confirmed the presence of 2PACz molecules on NiO_x surface. The high-resolution spectrum of the O 1*s* region elucidates the contribution of three different oxygen species. The fitted peak 1 (~529.2 eV), peak 2 (~530.8 eV), and peak 3 (~532.2 eV) are attributed to a typical Ni–O bond in NiO crystal, low-oxygen-coordinated defect site and the surface-adsorbed oxygen species, as well as interfacial interaction between Ni and O (from organic groups), respectively^{1,2}. After the deposition of SAM, the relative intensity of peak 3 apparently increases, suggesting a large amount of 2PACz molecules interact with NiO_x surface via the Ni–O interactions. In addition, the whole Ni 2*p* spectrum shifts towards a higher binding energy side by ~0.3 eV after the SAM treatment. The observed chemical shift suggests that the oxygen atoms in 2PACz are contributing lone pairs of electrons to the vacant orbitals of Ni atoms. As a result, coordinate covalent bonds of P=O···Ni are formed at the interface, as illustrated in Fig. 1a³.

In the Supplementary References section:

1. Alghamdi, A. R. M., Yanagida, M., Shirai, Y., Andersson, G. G. & Miyano, K. Surface passivation of sputtered NiO_x using a SAM interface layer to enhance the performance of perovskite solar cells. *ACS Omega* **7**, 12147–12157 (2022).
2. Cheng, M., Fan, H., Song, Y., Cui, Y. & Wang, R. Interconnected hierarchical NiCo₂O₄ microspheres as high-performance electrode materials for supercapacitors. *Dalton Trans.* **46**, 9201–9209 (2017).
3. Sun, J. et al. NiO_x-seeded self-assembled monolayers as highly hole-selective passivating contacts for efficient inverted perovskite solar cells. *Sol. RRL* **5**, 2100663 (2021).

In the Methods section:

‘XPS data were recorded on a Thermo Scientific ESCALAB 250Xi spectrometer under 2×10^{-10} Pa and X-ray source (XR6 monochromated Al K α , $h\nu = 1486.68$ eV) was exerted with a 500 μm spot size and 20 eV pass energy.’

Regarding the surface coverage of SAM, following the reviewer’s suggestion, we have conducted scanning electron microscope (SEM) and energy dispersive spectrometer (EDS) measurements to characterize ITO/NiO_x and ITO/NiO_x/2PACz samples. Unfortunately, due to the insufficient sensitivity for EDS to capture the information from a single layer, the acquired ‘signal’ is just basic noise instead of a real N and P distribution from 2PACz, as shown in Figure R1. Therefore, we consider this technique ineffective in characterizing the surface coverage of 2PACz properly.

Figure R1. SEM and EDS analysis of the NiO_x and NiO_x/SAM samples.

However, we could confirm that the surface coverage of SAM was decent via the KPFM characterization. After depositing SAM, there is a significant difference in the average surface potential between the NiO_x (0.588 V) and NiO_x/2PACz (0.185 V) samples. Through investigating the surface potential distribution of the selected area (Supplementary Figure 2, also shown below), we could find that there is no signal around 0.588 V from NiO_x (which would appear as pink dots in Supplementary Figure 2b) in the NiO_x/2PACz sample, indicating that the NiO_x was fully covered by the 2PACz monolayer. We therefore, added the following discussion in the revised manuscript.

On page 3:

‘Moreover, the potential mappings in both cases show that the films were homogeneous, reflecting that 2PACz was uniformly bonded onto the NiO_x surface with high coverage, which is also confirmed by the surface potential distribution results (Supplementary Figure 2).’

In Supplementary Information:

Supplementary Figure 2 | Distribution of surface potential of each pixel in Fig. 1c. a NiO_x surface and b NiO_x/SAM surface. In the NiO_x/SAM surface, all the pixels have surface potentials between

140–210 mV, which show in green color, and no pixel with a surface potential of 560–630 mV (in pink color) is observed. This result suggests that no NiO_x surface was exposed in the NiO_x/SAM film and SAM covers the NiO_x surface in high coverage.

2) The intermolecular interaction between PVK and carbazole monolayer was defined as van der Waals. What about pi-pi staking or dipole-dipole interaction? To me, these are better suitable than van der Waals. Describe more details about how it can be defined as van der Waals or provide the relevant evidence.

Reply:

We thank the reviewer for their comments. The interaction between carbazole units has been intensively investigated in the literature, especially in the organic optoelectronic field. The dipole-dipole interaction, pi-pi stacking and even van der Waals force were used to describe this interaction in different papers.

*For instance, to explore the interaction of carbazole donors and acceptors, the electrochemical cyclic voltammetric measurements were performed, and the results indicate that self-complementary dipole-dipole and π -stacking interactions between carbazole donors and acceptors existed (Dyes and Pigments 2020, **182**, 108474). These two supramolecular interactions lead to a smaller distance between the carbazole molecules (3.45 Å) than the general π -stacking distance (3.54 Å).*

*Also, the structural estimation could be performed by in-plane X-ray diffraction (XRD) and near-edge X-ray absorption fine structure spectroscopy (Polym. Adv. Technol. 2007, **18**, 353–363; Colloids and Surfaces A: Physicochem. Eng. Aspects 2008, **322**, 239–242). In these studies, the well-ordered arrangement of functional groups is caused by the enhancement of π - π interaction between the carbazole rings.*

*Moreover, it has also been reported that the nanostructure established by carbazole dendrons is primarily impacted by the increase in van der Waals forces with the increasing amount of carbazole units per dendron generation on a hydrophilic mica surface (Langmuir 2011, **27**, 9327–9336).*

Therefore, multi-interaction can exist between the carbazole units in SAM and PVK, and it is very difficult to access the physical information of a monolayer (especially with an amorphous PVK layer on top) by general characterization to determine the dominant interaction type. In such a case, van der Waals forces, which, broadly speaking, is used to describe the intermolecular interaction and hence involves the pi-pi staking and dipole-dipole interaction, is the best terminology to

comprehensively describe this complex case. To better clarify this point, we revised the following discussion in our revised manuscript.

On page 5:

‘In contrast, after introducing the SAM, the formation of covalent bonds between the NiO_x surface and phosphonic acid groups in 2PACz, as well as the formation of stronger intermolecular van der Waals interactions (including π - π stacking and dipole-dipole interaction, etc) between the 2PACz carbazole groups and the carbazole units in PVK (Fig. 2b), contribute to a more robust interface that alleviates the solvent washing effect to the PVK film^{20–23}.’

In References:

*20. Li, X., Wang, Y., Li, F. & Zhang, X. Fluorescent carbazole-containing dyes: synthesis and supramolecular assembly by self-complementary donor-acceptor π -stacking and dipolar interactions. *Dyes and Pigments* **182**, 108474 (2020).*

*21. Hoshizawa, H., Masuya, R., Masuko, T. & Fujimori, A. Control of orientation for carbazole group in comb copolymers arranged by method of organized molecular films. *Polym. Adv. Technol.* **18**, 353–363 (2007).*

*22. Hoshizawa, H. & Fujimori, A. Control of arrangement for carbazole group in fluorinated comb copolymers by method of organized films. *Colloids and Surfaces A: Physicochem. Eng. Aspects* **322**, 239–242 (2008).*

*23. Felipe, M. J. et al. Interfacial behavior of OEG-linear dendron monolayers: aggregation, nanostructuring, and electropolymerizability. *Langmuir* **27**, 9327–9336 (2011).*

3) A big concern to me is lack of discussion on charge transport across SAM although the central theme of this work is using SAM for hole injection layer. Charge transport across SAM can depend on the dipole of SAM (refs: *J. Phys. Chem. Lett.* 2018, 9, 17, 5078–5085; *J. Am. Chem. Soc.* 2019, 141, 22, 8969–8980). However, it has been also reported that the dipole of individual molecule can be canceled out in SAM (mostly done by Whitesides group; refs: *J. Am. Chem. Soc.* 2014, 136, 1, 16–19; *Angew. Chem. Int. Ed.* 2012, 51, 4658–4661; *J. Am. Chem. Soc.* 2015, 137, 11, 3852–3858; *JACS*, 2012, 134, 10876–10884). These discussions on the role of dipole on charge transport across SAM were expected but lacked. These thus must be added with the extensive reference citations.

Reply:

*We thank the reviewer for their comments. As stated by the reviewer, charge transport across SAM indeed depends on the dipole of SAM (*J. Phys. Chem. Lett.* 2018, **9**, 5078–5085; *J. Am. Chem. Soc.* 2019, **141**, 8969–8980). Here, the SAM is introduced to reduce the energy barrier for hole injection. However, as shown in Fig. 2h, the electrostatic potential step ($\Delta\Phi$) for charge transport across SAM*

has a large energy barrier (~ 1.7 eV), implying that the electron injection from NiO_x to 2PACz is difficult. Of course, as mentioned by the reviewer (*J. Am. Chem. Soc.* 2014, **136**, 16–19; *Angew. Chem. Int. Ed.* 2012, **51**, 4658–4661; *J. Am. Chem. Soc.* 2015, **137**, 3852–3858; *J. Am. Chem. Soc.* 2012, **134**, 10876–10884), it has a possibility that the electron tunnelling could lead to charge transport from NiO_x to 2PACz, while the tunnelling possibility is negatively proportion to the magnitude of the energy barrier for electron injection. So, provided such a large energy barrier for electron injection, the electron tunnelling possibility is low. Therefore, to better clarify this point, we revised the following discussion in our revised manuscript.

On page 6:

‘In addition, as shown in Fig. 2h, due to the electron transfer from NiO_x to 2PACz, the formed dipole at their interface led to an electrostatic potential step of +1.7 (–1.7) eV from NiO_x to 2PACz for the electron (hole), which reduced the energy barrier for hole injection and affected charge transport across SAM^{25,26}. Within such a large energy barrier for electron injection, the electron tunnelling possibility is relatively low^{27–30}.’

In References:

25. Chen, J. et al. Understanding Keesom interactions in monolayer-based large-area tunneling junctions. *J. Phys. Chem. Lett.* **9**, 5078–5085 (2018).

26. Baghbanzadeh, M. et al. Dipole-induced rectification across Ag^{TS}/SAM//Ga₂O₃/EGaIn junctions. *J. Am. Chem. Soc.* **141**, 8969–8980 (2019).

27. Yoon, H. J., Bowers, C. M., Baghbanzadeh, M. & Whitesides, G. M. The rate of charge tunneling is insensitive to polar terminal groups in self-assembled monolayers in Ag^{TS}S(CH₂)_nM(CH₂)_mT//Ga₂O₃/EGaIn junctions. *J. Am. Chem. Soc.* **136**, 16–19 (2014).

28. Yoon, H. J. et al. The rate of charge tunneling through self-assembled monolayers is insensitive to many functional group substitutions. *Angew. Chem. Int. Ed.* **51**, 4658–4661 (2012).

29. Liao, K.-C., Bowers, C. M., Yoon, H. J. & Whitesides, G. M. Fluorination, and tunneling across molecular junctions. *J. Am. Chem. Soc.* **137**, 3852–3858 (2015).

30. Thuo, M. M. et al. Replacing –CH₂CH₂– with –CONH– does not significantly change rates of charge transport through Ag^{TS}-SAM//Ga₂O₃/EGaIn junctions. *J. Am. Chem. Soc.* **134**, 10876–10884 (2012).

4) The following studies involving SAM in perovskite interface engineering are directly suitable to this work and thus should be cited: *Adv. Opt. Mater.* 2022, 10, 1, 2101361; *Adv. Energy Mater.* 2020, 10, 44, 2002606; *Nano Energy*, 2020, 73, 104752.

Reply:

We thank the reviewer for their suggestions. We agree that these important studies should be involved to improve the scientific quality of this manuscript. We have added the aforementioned references in the revised manuscript.

On page 3:

‘This SAM not only improved the robustness and energy level coupling between the NiO_x and PVK layers, but also provided a much-improved surface trap passivation effect to the NiO_x simultaneously 11–13.’

In References:

*11. Lee, J. Y., Kim, S. Y. & Yoon, H. J. Small molecule approach to passivate undercoordinated ions in perovskite light emitting diodes: progress and challenges. *Adv. Opt. Mater.* **10**, 2101361 (2022).*

*12. Kim, S. Y., Cho, S. J., Byeon, S. E., He, X. & Yoon, H. J. Self-assembled monolayers as interface engineering nanomaterials in perovskite solar cells. *Adv. Energy Mater.* **10**, 2002606 (2020).*

*13. Mak, C. H. et al. Recent progress in surface modification and interfacial engineering for high-performance perovskite light-emitting diodes. *Nano Energy* **73**, 104752 (2020).*

5) Comparison of surface potential using KPFM is often misleading as the reference is missing.

Reply:

We thank the reviewer for their reminder.

We agree with the reviewer that the work function of KPFM tip (φ_{tip}), which serves as a reference, should be mentioned to enable valid comparison of surface potential data between different samples. In our experiment, both samples were measured by the same KPFM tip with a work function of 4.42 eV. Based on the measured surface potential (V_{CPD}) of 0.588 V and 0.185 V for the NiO_x sample and NiO_x/SAM sample, respectively, we can calculate the work function of the samples (φ_{sample}) as 3.832 eV and 4.235 eV for the NiO_x and NiO_x/SAM, respectively, as per the following equation:

$$V_{\text{CPD}} = \frac{\varphi_{\text{tip}} - \varphi_{\text{sample}}}{e}$$

This result is consistent with our DFT calculation that a deeper work function is achieved by incorporating the SAM. Also, the work function increases by ~0.4 eV after SAM treatment also matches well with the HOMO level increase as shown in Fig. 3b.

To better clarify the KPFM measurement and the acquired results, we revised the following discussion and added more information about the KPFM in our revised manuscript.

On page 3:

‘KPFM study also reveals that the surface potential of 0.588 V (corresponding to a work function of 3.832 eV) of the pristine NiO_x had dramatically decreased to 0.185 V (corresponding to a work function of 4.235 eV) when the SAM was formed, suggesting an increase of work function by ~0.4 eV for the NiO_x/SAM sample.’

In the Methods section:

‘In our experiment, both samples were measured by the same KPFM reference tip with a work function of 4.42 eV. Along with the surface potential (V_{CPD}), the work function of the samples (φ_{sample}) can be calculated by equation of $V_{CPD} = \frac{\varphi_{tip} - \varphi_{sample}}{e}$.’

6) Figure 2b means a lot. First, it means carbazole terminal groups are loosely packed (i.e., SAM is disordered largely) otherwise the counterpart carbazole substituent from PVK polymer cannot be intercalated as illustrated. Second, the loosely packed SAM is free from the dipole cancellation maximizing the dipole effect on WF. However, I cannot find any evidence or justification that supports the Figure 2b. Is this because of the surface of NiO_x substrate is rough causing the disorder of SAM? Or, the SAM is prepared by spin-coating for a short time (30 seconds), which can give sparse SAM rather than dense one, which is not sure intended or not.

Reply:

We thank the reviewer for their comments.

The initial version of Fig. 2b presented an idealized schematic diagram illustrating how 2PACz and PVK pack together via their carbazole groups. We concur with the reviewer's assessment that we lack enough evidence to substantiate this ideal packing model in a real-world context.

Given the ultra-thin nature of SAM and the disordered configuration of the polymer, it poses significant challenges to pinpoint the exact interfacial arrangement between a monolayer and a polymer. However, from our mechanical peeling experiment, we managed to affirm the existence of robust intermolecular interaction between SAM and PVK. More details on this can be found in our response to Point 10 from Reviewer #2. Despite this, the detailed packing configuration remains elusive.

In light of the above, we have revised Fig. 2b in our manuscript (also shown below) to depict a more plausible representation of the random packing scenario between SAM and PVK. We are confident that this revised diagram better reflects the range of possible packing configurations between these two components, bringing us closer to portraying their potential interaction.

On page 7:

Fig. 2 Impact of PVK adhesion. b The schematic diagram showing the detailed configuration of NiO_x/SAM/PVK resulted in a robust interface.

Reviewer #2 (Remarks to the Author):

In this manuscript, Zhenchao Li et al. introduced a SAM layer between NiO_x and PVK to build a robust interface, passivate interfacial deep traps, and align the energy levels. Their as-fabricated blue and green PeLEDs with external quantum efficiencies of 14.5% and 26.0%, respectively. The SAM layer also enhances the devices' response speed. Overall, this work is impressive and interesting. I found some major issues in these tests and analyses, as described below.

Reply:

We thank the reviewer for their critical and valuable comments.

We have made a major revision to further improve the scientific quality of this manuscript. And we hope the reviewer finds the revised manuscript suitable for publication in Nature Communications.

1. There are a lot of available SAM layers, such as 2PACz, 3PACz, 4PACz. Why are the authors focused on 2PACz? The comparison among various SAM layers are recommended to enrich this manuscript.

Reply:

We thank the reviewer for their comments.

During our research, we have investigated the potential of several types of SAMs, including carbazole-based SAMs (i.e., 2PACz and MeO-2PACz) and alkyl-chain-based SAMs [i.e., ethylphosphonic acid (EPA) and 2-Aminoethylphosphonic acid (AEPA)], as shown in Figure R2. We explored their impact on device performances of blue and green PeLEDs using the same device architecture of ITO/NiO_x/SAM/PVK/blue or green perovskite/TPBi/LiF/Al. According to Figure R2, all the SAMs can reduce the turn-on voltage of the devices, compared to those without SAM. Among them, 2PACz exhibited the highest EQE in both blue and green devices, possibly due to the molecular identity of the carbazole unit between 2PACz and PVK. Table R1 shows that the best EQEs of blue devices fabricated with 2PACz, MeO-2PACz, EPA and AEPA were 14.5%, 8.8%, 5.4% and 0.8%, respectively. While the best EQEs of green devices fabricated with 2PACz, MeO-2PACz, EPA and AEPA were 26.0%, 19.3%, 17.7% and 2.0%, respectively. Consequently, we specifically focus our research on 2PACz as which provides us with the best device performance and probably has more interesting science inside.

Figure R2. Device performance of the blue and green PeLEDs using different SAM layers. *a* Molecular structures of the 2PACz, MeO-2PACz, ethylphosphonic acid (EPA) and 2-

Aminoethylphosphonic acid (AEPA), respectively. Current density–voltage curves (dotted lines) and luminance–voltage curves (solid lines) of the (b) blue and (d) green PeLEDs with various SAM layers. Current density–EQE curves of the (c) blue and (e) green PeLEDs with various SAM layers.

Table R1. Summary of device performances of blue and green PeLEDs with various SAMs.

SAM	V_T for blue (V)	Max EQE for blue (%)	V_T for green (V)	Max EQE for green (%)
2PACz	2.2	14.5	2.2	26.0
MeO-2PACz	2.2	8.8	2.2	19.3
EPA	2.4	5.4	2.2	17.7
AEPA	2.8	0.8	2.4	2.0

We strongly agree with the reviewer that the comparison among various SAMs would further improve the scientific quality of this manuscript. However, as the EPA and AEPA are based on an alkyl-chain-based system which is different from the carbazole-based system studied in our research, in-depth science is probably different between these two systems and deserves more effort to provide a clear picture. In such a case, to provide a more systematic study of the impact of the carbazole unit, we only added the discussion and data of MeO-2PACz in our revised manuscript.

On page 9:

‘We also consider that the identity of the carbazole unit in both SAM and PVK might be critical for device performance improvement as when we changed 2PACz to MeO-2PACz that involves two more –OCH₃ groups in the carbazole unit, a poorer device performance was achieved (Supplementary Figure 14).’

In Supporting Information:

Supplementary Figure 14 | Device performance of the blue and green PeLEDs with device architecture of ITO/NiO_x/MeO-2PACz/PVK/Perovskite/TPBi/LiF/Al. a Current density–voltage curves (dotted lines) and luminance–voltage curves (solid lines) of the blue and green PeLEDs with MeO-2PACz layer. **b** Current density–EQE curves of the blue and green PeLEDs with MeO-2PACz layer. The best EQEs for blue and green PeLEDs are 8.8% and 19.3%, respectively. Inset is the chemical structure of the MeO-2PACz molecule.

Regarding the investigation of other SAMs like 3PACz and 4PACz, we highly appreciate the reviewer’s constructive reminder. We also believe after studying a set of SAMs with various chain lengths can provide us with systematic results and allow us to comprehensively understand the impact of chain length on the interfacial dipole formation, packing configuration between SAMs and PVK, as well as physical adhesion between SAMs and PVK, etc. We consider this impressive topic deserves a new project to fully explore, which will definitely be one of our main research directions. Therefore, we decided not to involve this content in our revised manuscript which is already very informative.

2. Why choose the Al electrode for perovskite devices? Will Al ions migrate through the device and react with perovskites?

Reply:

*We thank the reviewer for their comments. In the PeLED field, LiF/Al electrode is widely used for electron injection because of the lower work function. The chemical reaction between LiF and Al can reduce the work function of Al to meet this end (Appl. Phys. Lett. 2009, **94**, 063302), while LiF cannot react with other metals like Ag and Au to have a similar effect (i.e., reducing work function). Therefore, we added the following discussion to clarify the reason why the LiF/Al electrode was selected.*

In the Methods section:

‘LiF/Al electrode is selected because the chemical reaction between LiF and Al (forming chemical bonds of Al–F at the interface), can reduce the work function of this electrode to ~ 2.9 eV ⁷⁴.’

In References:

74. Xie, Z. T. et al. Interfacial reactions at Al/LiF and LiF/Al. *Appl. Phys. Lett.* **94**, 063302 (2009).

In relation to the potential migration of Al towards the perovskite layer, our previous study (published in Nature Communications, 2019, 10, 1027) provides some insight. According to our findings, a clear boundary exists between the Al electrode and TPBi, as depicted in Figure R3. This observation implies that there is no diffusion of Al occurring within this particular device.

Figure R3. STEM image of the perovskite light-emitting diode with a configuration of ITO/PEDOT:PSS/Perovskite/TPBi/LiF/Al. (*Nat. Commun.* 2019, 10, 1027)

3. “After cooling to room temperature, the NiO_x-coated substrates were subjected to a 5-min oxygen plasma treatment”. Why is the oxygen plasma treatment needed here?

Reply:

We thank the reviewer for their comments.

In our previous work (Joule 2021, 5, 456–466), this method has been used to prepare the NiO_x layer according to the literature (Sol. RRL 2019, 3, 1900045). The plasma treatment increases the surface wettability, electrical conductivity, and work function of the prepared NiO_x film, which benefits the device performance. According to the literature, the same approach was also applied to

the fabrication of high-efficiency green PeLEDs that the glass/ITO/NiO_x or Ni_{1-y}Mg_yO_x substrates were treated with oxygen plasma for 30 minutes before the spin-coating of perovskite precursor solution (Adv. Mater. 2023, DOI: 10.1002/adma.202302283). To better clarify this point, we added the following discussion in our revised manuscript.

In the Methods section:

‘After a 5-min oxygen plasma treatment of ITO substrates, the NiO_x layers were deposited onto these ITO substrates by spin-casting the 0.45 μm PTFE-filtered NiO_x precursor solution at 3000 rpm for 50 s and then baking them at 270 °C for 45 min in ambient air. After cooling to room temperature, the NiO_x-coated substrates were subjected to another 5-min oxygen plasma treatment to modify their surface properties (e.g., increasing the surface wettability, electrical conductivity, and work function)^{72,73}.’

In References:

72. Wang, T. et al. Efficient inverted planar perovskite solar cells using ultraviolet/ozone-treated NiO_x as the hole transport layer. Sol. RRL 3, 1900045 (2019).

73. Bai, W. et al. Perovskite light-emitting diodes with an external quantum efficiency exceeding 30%. Adv. Mater. (2023). <https://doi.org/10.1002/adma.202302283>

4. The photos for the blue and green PeLED devices are needed in the supplementary information.

Reply:

We thank the reviewer for their reminder. We have inserted the photos of the blue and green devices in the revised Supplementary Figures 15 and 17, respectively, which are also shown below.

In Supplementary Information:

Supplementary Figure 15 | Spectrum stability of blue PeLEDs with an emission peak at ~493 nm. a EL spectra of the device without SAM under an applied voltage ranging from 3.8 V to 7 V. **b** EL spectra of the device with SAM under an applied voltage ranging from 2.1 V to 7 V. Inset is the photo of a working blue PeLED. Normalized EL spectra of the device **(c)** without SAM and **(d)** with SAM under continuous operation (at the max EQE). No spectrum shift was observed in any case.

Supplementary Figure 17 | Spectrum stability of green PeLEDs with an emission peak at ~515 nm. a EL spectra of the device without SAM under an applied voltage ranging from 2.7 V to 7 V. **b**

EL spectra of the device with SAM under an applied voltage ranging from 2.1 V to 7 V. Inset is the photo of a working green PeLED.

5. For Supplementary Figure 5, what caused the difference in the positions of the PL emission peaks?

Reply:

We thank the reviewer for their comments.

In our previous work, we concluded that different substrates had a negligible effect on the formation of the quasi-2D perovskite phase. This conclusion was based on the observation that the peak positions in the X-ray diffraction (XRD) signals, corresponding to both the 2D and the 3D perovskite regions, remained unchanged. However, upon the reviewer's suggestion, we further examined the XRD signal intensities of the 2D and the 3D phases.

We found a slight increase in the intensity ratio of the 3D to the 2D perovskite peaks as the substrates changed from NiO_x to NiO_x/PVK, and finally to NiO_x/SAM/PVK (Supplementary Figures 9b and 9c, also shown below). This suggested a higher concentration of the 3D phase in the perovskite film formed on the NiO_x/SAM/PVK substrate.

Given the efficient energy and/or charge transfer from the 2D to the 3D phase in quasi-2D perovskite systems, this slight increase in the 3D phase could explain the observed red-shift in the photoluminescence (PL) spectra of perovskite films when the substrates change from NiO_x to NiO_x/PVK, and finally to NiO_x/SAM/PVK.

To explain this subtle PL red-shift more clearly, we have revised Supplementary Figure 9 and the associated legend in Supplementary Figure 8 of our manuscript.

In Supplementary Information:

Supplementary Figure 9 | Film quality of (PEA/IPA)₂Cs_{n-1}Pb_nBr_{3n+1} perovskite on various substrates. a Morphologies, Left panel: SEM images; Right panel: AFM images. In both cases, the

perovskite film morphologies are similar, suggesting that introducing the 2PACz SAM had a negligible impact on the perovskite morphology. **b** X-ray diffraction patterns. The similar X-ray diffraction peak positions in all cases suggest that different substrates had a negligible impact on forming the quasi-2D perovskite phase. However, the perovskite formed at NiO_x substrate had a poorer crystallinity. **c** The intensity ratio between the 3D and 2D perovskite peaks shown in **b**. An increasing 3D perovskite phase is found when changing the substrates from NiO_x to NiO_x/PVK to NiO_x/SAM/PVK.

Supplementary Figure 8 | PL properties of (PEA/IPA)₂Cs_{n-1}Pb_nBr_{3n+1} perovskite films on NiO_x, NiO_x/PVK and NiO_x/SAM/PVK substrates. a Photos of perovskite films under 365-nm ultraviolet lamp excitation. **b** PL spectra of the corresponding perovskite films. The slight red shift of PL from NiO_x to NiO_x/PVK to NiO_x/SAM/PVK cases is due to the increasing 3D perovskite ratio in quasi-2D perovskite films, according to the XRD results shown in Supplementary Figure 9.

6. For the device stability test (Supplementary Figure 12), what are the test conditions (temperature, humidity...)? This important information should be disclosed in the Methods section so that other groups can reproduce the authors' results.

Reply:

The stability test was carried out in a N₂-filled glovebox without encapsulation (25 °C, H₂O concentration lower than 0.1 ppm). We will disclose this information in the Methods in the revised manuscript.

In the Methods section:

‘The stability test was carried out in a N₂-filled glovebox without encapsulation (25 °C, H₂O concentration lower than 0.1 ppm).’

7. Although SAM enhances the devices’ lifetime, it is still relatively short (<1 h). Have the authors done any encapsulation for the as-fabricated PeLEDs? The short lifetime prevents PeLED from being a real application. Are there any other factors limiting devices’ lifetime?

Reply:

We thank the reviewer for their comments.

We concur with the reviewer that device lifetime is a critical parameter in assessing device quality. Hence, it's vital to discuss more clearly the factors that constrain this lifetime.

*Initially, we measured the device lifetime inside a nitrogen-filled glovebox, which meant the device wasn't encapsulated. Additionally, we based the blue perovskite formula and device architecture on existing literature (Nat. Commun. 2018, **9**, 3541). According to this source, the device's intrinsic lifetime is a mere 60 seconds. However, we managed to enhance this to 300 seconds in our reference device (i.e., the one without SAM), an improvement of five times over the reported lifetime. Despite this enhancement, we agree with the reviewer that this duration could still be deemed short, reflecting what we believe to be the inherent limitation of this material system.*

Regarding our PeLED device, aside from the factor we discussed in our manuscript (i.e., the impact from the organic/inorganic heterointerfaces), we believe our devices are also affected by other general factors contributing to device degradation. These include:

*1. Electric-field-induced ion migration and phase segregation, which can dismantle perovskite lattices, generate defects, corrode electrodes, degrade charge transport layers, and form charge-accumulated interfaces (Nat. Commun. 2018, **9**, 3541; Adv. Energy Mater. 2016, **6**, 1502246).*

*2. Electrical stress, which similarly leads to ion migration (ACS Appl. Mater. Interfaces 2016, **8**, 5351–5357; Nat. Commun. 2016, **7**, 13422).*

*3. Joule heat, which degrades the perovskite crystals and functional layers (Synthetic Metals 2016, **216**, 40–50; Adv. Funct. Mater. 2021, **31**, 2103219).*

For clarity, we've revised the caption in Supplementary Figure 16.

Supplementary Figure 16 | Device lifetime of the best blue PeLEDs with and without SAM.

Besides the effect of the organic/inorganic heterointerfaces mentioned in the main text, electric-field-induced ion migration and phase segregation, electrical stress, and Joule heat are also considered the factors that lead to the degradation of our devices^{4–9}.

In the Supplementary References section:

4. Xing, J. et al. Color-stable highly luminescent sky-blue perovskite light-emitting diodes. *Nat. Commun.* **9**, 3541 (2018).

5. Carrillo, J. et al. Ionic reactivity at contacts and aging of methylammonium lead triiodide perovskite solar cells. *Adv. Energy Mater.* **6**, 1502246 (2016).

6. Chen, S. et al. Mobile ion induced slow carrier dynamics in organic–inorganic perovskite CH₃NH₃PbBr₃. *ACS Appl. Mater. Interfaces* **8**, 5351–5357 (2016).

7. Ahn, N. et al. Trapped charge-driven degradation of perovskite solar cells. *Nat. Commun.* **7**, 13422 (2016).

8. Tyagi, P., Srivastava, R., Indu Giri, L., Tuli, S. & Lee C. Degradation of organic light emitting diode: heat related issues and solutions. *Synthetic Metals* **216**, 40–50 (2016).

9. Zou, G. et al. Color-stable deep-blue perovskite light-emitting diodes based on organotrichlorosilane post-treatment. *Adv. Funct. Mater.* **31**, 2103219 (2021).

8. A summary table on the lifetime of reported PeLED is suggested. We should pay attention to devices' stability just like perovskite solar cell field.

Reply:

We thank the reviewer for their comments. We agree with the reviewer that device stability is a key issue in this field, which limits the development of blue PeLEDs and their applications. We have

supplemented a summary table focusing on the blue PeLEDs in the Supporting Information section, as shown below. By comparing other results, we find that our device lifetime is at a moderate level.

On page 9:

‘The half-lifetime of the device increases from 300 s to 831 s after SAM modification (Supplementary Figure 16), which is at a moderate level compared with other published results (Supplementary Table 1).’

In Supporting Information:

Supplementary Table 1 | Summary of stability of the blue PeLEDs in the past 4 years.

Perovskite composition	EL peak (nm)	Testing mode	Testing condition	Lifetime	Ref.
CsPbBr ₃ QDs	480	Constant current density	L ₀ =100 cd m ⁻²	126 min	[10]
(PCTA) ₂ CsPb ₂ Br ₇	480	Constant current density	L ₀ =67 cd m ⁻² J ₀ =1 mA cm ⁻²	21.6 min	[11]
CsPbBr _{3-x} Cl _x + CsFA-Ac	477	Constant current density	L ₀ =100 cd m ⁻²	120 s	[12]
PEA ₂ (Rb _x Cs _{1-x}) _{n-1} Pb _n (Br _{1-y} Cl _y) _{3n+1}	475	Constant bias	L ₀ =100 cd m ⁻² V ₀ =3.5 V	100 s	[13]
CsPbBr ₃ QDs	469	Constant current density	L ₀ =115 cd m ⁻² J ₀ =12.5 mA cm ⁻²	25 h	[14]
PEA ₂ (Cs _{1-x} EA _x PbBr ₃) ₂ PbBr ₄	488	Constant current density	L ₀ =100 cd m ⁻² J ₀ =1.5 mA cm ⁻²	1h	[15]
PEA ₂ Cs _{n-1} Pb _n (Cl _x Br _{1-x}) _{3n+1}	480	Constant bias	V ₀ =4.4 V	10 min	[16]
PEACl + CsPbBr ₃ + YCl ₃	485	Constant bias	L ₀ =100 cd m ⁻² V ₀ =3.2 V	80 min	[17]
PEA ₂ Cs _{2-x} EA _x Pb ₃ Br ₁₀	490	Constant current density	L ₀ =60 cd m ⁻² J ₀ =0.44 mA cm ⁻²	55.3 min	[18]
(Cs/Rb/K/PEA)Pb(Br/Cl) ₃	488	Constant current density	L ₀ =100 cd m ⁻²	5.12 min	[19]
CsPb(Cl/Br) ₃	487	Constant current density	L ₀ =178 cd m ⁻² J ₀ =1 mA cm ⁻²	2900 s	[20]
p-F-PEA ₂ Cs _{n-1} Pb _n (Cl _x Br _{1-x}) _{3n+1}	489	Constant current density	J ₀ =1 mA cm ⁻²	10.4 min	[21]
(PEA/IPA) ₂ Cs _{n-1} Pb _n Br _{3n+1}	493	Constant current density	L ₀ =100 cd m ⁻² J ₀ =1 mA cm ⁻²	~14 min	This work

In the Supplementary References section:

- Jiang, Y. et al. Synthesis-on-substrate of quantum dot solids. *Nature* **612**, 679–684 (2022).
- Zhou, Y.-H. et al. Spectral stable blue perovskite light-emitting diodes by introducing organometallic ligand. *Adv. Opt. Mater.* **10**, 2101655 (2022).
- Ding, W. et al. Transformation of quasi-2D perovskite into 3D perovskite using formamidine acetate additive for efficient blue light-emitting diodes. *Adv. Funct. Mater.* **32**, 2105164 (2022).

13. Yang, Y. et al. Highly efficient pure-blue light-emitting diodes based on rubidium and chlorine alloyed metal halide perovskite. *Adv. Mater.* **33**, 2100783 (2021).
14. Bi, C. et al. Suppressing auger recombination of perovskite quantum dots for efficient pure-blue light-emitting diodes. *ACS Energy Lett.* **8**, 731–739 (2023).
15. Chu, Z. et al. Large cation ethylammonium incorporated perovskite for efficient and spectra stable blue light-emitting diodes. *Nat. Commun.* **11**, 4165 (2020).
16. Li, Z. et al. Modulation of recombination zone position for quasi-two-dimensional blue perovskite light-emitting diodes with efficiency exceeding 5%. *Nat. Commun.* **10**, 1027 (2019).
17. Wang, Q. et al. Efficient sky-blue perovskite light-emitting diodes via photoluminescence enhancement. *Nat. Commun.* **10**, 5633 (2019).
18. Liu, S. et al. Zwitterions narrow distribution of perovskite quantum wells for blue light-emitting diodes with efficiency exceeding 15%. *Adv. Mater.* **35**, 2208078 (2022).
19. Zhu, C. et al. High triplet energy level molecule enables highly efficient sky-blue perovskite light-emitting diodes. *J. Phys. Chem. Lett.* **12**, 11723–11729 (2021).
20. Shen, Y. et al. Multifunctional crystal regulation enables efficient and stable sky-blue perovskite light-emitting diodes. *Adv. Funct. Mater.* **32**, 2206574 (2022).
21. Xia, Y. et al. Reduced confinement effect by isocyanate passivation for efficient sky-blue perovskite light-emitting diodes. *Adv. Funct. Mater.* **32**, 2208538 (2022).

9. The SAM also improves the response speed of PeLED. However, statistical data is needed to reach a reliable conclusion.

Reply:

We thank the reviewer for their reminder. We agree with the reviewer that presenting statistical data can consolidate our conclusion that the response time of the PeLEDs could be decreased with the modification of SAM. Therefore, we collected the response time from 10 devices, and the results exhibited a low level of deviation (Supplementary Figure 18, also shown below). Therefore, we added the following discussion and figure in the main text and Supporting Information.

On page 11:

‘Figure 4a shows that the overall response times of the devices with and without SAM were 373 μ s and 540 μ s, respectively, demonstrating that the introduction of the SAM accelerated the device response, which is further confirmed by the statistic results (Supplementary Figure 18).’

In Supporting Information:

Supplementary Figure 18 | Statistical data of T_{rise} and T_{fall} under 100, 500, 1000, 2500, 5000 and 10000 Hz of the PeLEDs with and without SAM modification. The IQR represents interquartile range.

10. Is it possible to measure the mechanical adhesion of the device with and without a SAM layer?

Reply:

We appreciate the reviewer's constructive suggestion.

The mechanical adhesion measurement, as suggested by the reviewer, can indeed strengthen the conclusions drawn in our manuscript. To investigate this, we fabricated samples with and without SAM as shown in Supplementary Figure 7a. We then measured the force applied as a function of displacement in 10 samples using a universal tensile testing machine.

The average value of the force needed to peel off the PVK layer was determined during the stable process of layer detachment, as illustrated in Supplementary Figures 7b and 7c (also shown below). Our findings reveal that peeling off the PVK layer from a SAM substrate requires more force compared to a NiO_x substrate. This indicates that the 2PACz layer indeed enhances the PVK layer's adhesion strength (Supplementary Figure 7d).

We observed the adhesion strength between the PVK/SAM interface to be 14% stronger than that of the PVK/ NiO_x interface. Consequently, to reinforce our conclusion that the strong van der Waals force between PVK and SAM enhances the PVK layer's adhesion, we've incorporated this discussion and corresponding figure into the main text and Supporting Information, respectively.

On page 5:

‘To further examine the mechanical adhesion strength of the PVK/SAM and PVK/NiO_x interfaces, we carried out a series of measurements as depicted in Supplementary Figure 7. Our results indicate a greater force is required to detach the PVK layer from the SAM substrate as compared to the NiO_x substrate. This finding suggests that the 2PACz layer significantly bolsters the adhesion strength of the PVK layer. Additionally, we observed that the average adhesion strength at the PVK/SAM interface is approximately 14% stronger than at the PVK/NiO_x interface.’

In the Methods section:

‘The tensile test was based on a tensile machine developed by Dongriyiqi (DR-509AQ). The samples were prepared with a structure of Glass (15×15 mm)/ITO/NiO_x/SAM/PVK. Then one end of a 3M scotch tape (width of 12.7 mm) was firmly stuck on the surface of PVK. The other end of the tape was fixed on the up holder of the tensile machine for pulling up. Then a double-side tape was stuck to the other side of the sample and firmly mounted on the bottom holder. During pulling up of the up holder, the force-displacement relationship was detected by highly sensitive force and spatial sensors, respectively, and recorded by a computer.’

In Supporting Information:

Supplementary Figure 7 | The adhesion strength measurement between the PVK/SAM and PVK/NiO_x interfaces. a The schematic figure of the mechanical adhesion strength test. The force-

displacement curves of (b) with SAM and (c) without SAM layer. The necessary force to peel off the PVK layer is defined as the average value of the plateau of the curve (marked in color background), demonstrating the stable process during the layer peeling off. d The statistical data of the calculated adhesive strength of 10 samples (with and without SAM). The adhesion strength is calculated via dividing the necessary force by the width of the tape (12.7 mm). The error bars show the highest and lowest adhesive strength values for samples without and with SAM.

11. What are the breaking down voltages for blue and green PeLEDs?

Reply:

We thank the reviewer for reminding us to show the breaking-down voltage of our devices, which is another important parameter to study the device quality. In Fig. 3 of our revised manuscript (also shown below), we have shown data with extended voltage to 8 V (for blue PeLEDs) and 9 V (for green PeLEDs). We can find that for the blue devices, the maximum luminance of the device reached 10392 cd m⁻² at 6.6 V and 3888 cd m⁻² at 7.6 V for devices with and without SAM, respectively. A further increased applied voltage caused a decreased luminance, suggesting the breaking down of the device. Therefore, the breaking-down voltages for blue devices with and without SAM are 6.6 V and 7.6 V, respectively. Similarly, the breaking-down voltages for green devices with and without SAM are 7.2 V and 8.8 V, respectively.

According to the above results, the devices with SAM seem to have poorer device stability as they possess smaller breaking-down voltages. However, we should note that the current density of the devices with SAM is much higher than that of those without SAM under the same applied bias. Actually, under the breaking-down voltage of each device, their current densities are all around the scale of $\sim 10^2$ mA cm⁻² (176.2 mA cm⁻² and 54.0 mA cm⁻² for blue PeLEDs with and without SAM, respectively; 372.9 mA cm⁻² and 286.8 mA cm⁻² for green PeLEDs with and without SAM, respectively). Therefore, we consider that when devices reach a high current density of $\sim 10^2$ mA cm⁻², the generated Joule heat destroys the perovskite layers and hence causes the breaking down of the devices. Therefore, we revised Figures 3d and 3g and added the following discussion in our revised manuscript.

On page 9:

'It is interesting to find that all the devices started to break down at a current density on a scale of $\sim 10^2$ mA cm⁻². This corresponded to breakdown voltages of 6.6 V, 7.6 V, 7.2 V, and 8.8 V for blue devices with and without SAM, and green devices with and without SAM, respectively. We attribute

this phenomenon to the generation of Joule heat under such high current densities, which destroyed the perovskite layers and caused the devices to break down.'

Fig. 3 Current density–voltage curves (dotted lines) and luminance–voltage curves (solid lines) of the best (d) blue and (g) green PeLEDs with and without SAM.

12. How do the authors measure the EQE and luminescence of PeLEDs? The calibration step is important for the accuracy of results.

Reply:

The current density–voltage and luminance–voltage curves, EL spectra, and EQE of the PeLED were recorded simultaneously on a commercial system (XPQY-EQE-350-1100, Guangzhou Xi Pu Optoelectronics Technology Co., Ltd., China) that was equipped with an integrating sphere (GPS-4P-SL, Labsphere) and a photodetector array (S7031-1006, Hamamatsu Photonics).

We carried out the calibration via a standard lamp that was calibrated in the Shanghai standardization centre regularly. The photon number and photon energy were carefully aligned at both labs. The absolute spectral radiant flux is calibrated by using a NIST-traceable radiant-flux standard lamp (HL-3P-INT-CAL, Ocean Optics Co., Ltd.). The same measurement and calibration can also be found in other literature using the same instrument (Adv. Mater. 2023, 10.1002/adma.202302283). To make our description clearer, we revised the following description in the Methods section:

'The current density–voltage and luminance–voltage curves, EL spectra, and EQE of the PeLEDs were recorded simultaneously on a commercial system (XPQY-EQE-350-1100, Guangzhou Xi Pu Optoelectronics Technology Co., Ltd., China) that was equipped with an integrating sphere (GPS-

4P-SL, Labsphere) and a photodetector array (S7031-1006, Hamamatsu Photonics). Device operating lifetime was also measured by this system. The absolute spectral radiant flux is calibrated by using a NIST-traceable radiant-flux standard lamp (HL-3P-INT-CAL, Ocean Optics Co., Ltd.). The same measurement and calibration can also be found in other literature using the same instrument⁷³.

In References:

73. Bai, W. et al. Perovskite light-emitting diodes with an external quantum efficiency exceeding 30%. *Adv. Mater.* (2023). <https://doi.org/10.1002/adma.202302283>

Based on these main concerns, I think the quality of this manuscript is not good enough so far. I recommend further consideration for publication of this manuscript in *Nature Communications* after major revision.

Reply:

We thank the reviewer for their critical and valuable comments.

We have made a major revision to further improve the scientific quality of this manuscript. And we hope the reviewer finds the revised manuscript suitable for publication in Nature Communications.

Reviewer #3 (Remarks to the Author):

This manuscript reports on an approach to fabricate high-efficiency and fast-response perovskite LED by inserting a self-assembled monolayer (SAM) of [2-(9H-carbazol-9-yl)ethyl]phosphonic acid (2PACz) between NiOx and poly(9-vinylcarbazole) (PVK) layers. SAM improves robustness between NiOx and PVK layers, tunes energy level and passivates surface traps of NiOx layer. Perovskite LEDs using the tri-layer of NiOx/SAM/PVK have high EQE and luminance for both blue and green emissions, and also have significantly decreased turn-on voltage to 2.1 V and fast response time. Overall this is an interesting work but there are several important points that need to be addressed first. Detailed comments are as follows.

Reply:

We thank the reviewer for their very positive comments. We have made the necessary revisions to further improve the scientific quality of this manuscript.

1. In Fig. 1c, surface potentials of NiO_x and NiO_x/SAM layers from KPFM look similar and uniform in the current scale. Narrowing the scale bar would be helpful for comparing the surface potential and morphology more clear.

Reply:

We agree with the reviewer that showing the KPFM images in the same scale is not the best way to compare the difference between NiO_x and NiO_x/SAM layers, especially when their surface potentials are quite different. To provide a clearer comparison, as suggested by the reviewer, we narrow down the scale bar ranging from 560 mV to 630 mV for NiO_x surface and 140 mV to 210 mV for NiO_x/SAM surface, and the result is shown in the revised Fig. 1c (also attached below). The distribution of the surface potential is still uniform in both cases. We further analysed the KPFM data via NanoScope Analysis software. It shows that the surface potential of the selected area is in Gaussian distribution and the full width at half maxima is as small as ~28 mV in both cases (Figure R4), which also suggests that the surface potential distribution is narrow in the selected area.

To compare the surface potential and morphology clearer, we narrowed the scalebar for each sample and replaced the original Fig. 1c in the revised manuscript, as shown below:

Fig. 1 Changes in the surface after the formation of the 2PACz SAM. **a** The molecular structures of PVK and 2PACz, and a schematic diagram showing how the 2PACz molecules are bonded on the NiO_x surface via tridentate bonds, i.e., one coordination bond (via Lewis base–Lewis acid interaction with the Ni atom) and two covalent bonds (via dehydration condensation reactions with the hydroxyl

groups)¹⁷. **b** Contact angle measurements of NiO_x and NiO_x/SAM films. **c** Surface potential measurements of the NiO_x and NiO_x/SAM films by KPFM.

Figure R4. Distribution of surface potential of each pixel in Fig. 1c. (a) NiO_x surface and (b) NiO_x/SAM surface.

2. In Fig. 2b, PVK has strong interaction with 2PACz, so PVK can remain more than without 2PACz after the DMSO washing. However, there is no direct evidence for the strong interaction between PVK and 2PACz layers. So, adding the crystallinity or electrical property would be nice to support the robustness or strong interaction between 2PACz and PVK layers.

Reply:

We concur with the reviewer that characterizing the interaction between PVK and 2PACz layers would strongly support our claim. Indeed, we acknowledge that our original characterizations, such as absorption and contact angle measurements, provide somewhat indirect evidence.

We have also considered the characterizations suggested by the reviewer. However, the local packing of SAM and PVK is difficult to characterize, and the crystallinity of PVK is also difficult to accurately measure due to its amorphous nature. Regarding the electrical property of the multi-layer HTL itself, we conducted a measurement in our original paper (Fig. 3c) that the incorporation of SAM improved the electrical property of the HTL. However, it is still difficult to conclude that the effect is solely caused by the carbazole packing between SAM and PVK as the HOMO level changes and the interfacial dipole is involved, as discussed in the main text. Therefore, we consider these characterizations might still not be 'direct' enough to reveal the mechanical adhesion between PVK and 2PACz.

In such a case, along with the concerns raised by Reviewer #2 (Point 10), we conducted the mechanical peeling experiment, which served as a straightforward approach to unveil the mechanical adhesion strength between the PVK/SAM and PVK/NiO_x interfaces. Details of this result can be found in the answer to this next point.

3. To claim “better adhesion” between the SAM and PVK layer, mechanical peeling experiment should be performed.

Reply:

We appreciate the reviewer's constructive suggestion.

The mechanical adhesion measurement, as suggested by the reviewer, can indeed strengthen the conclusions drawn in our manuscript. To investigate this, we fabricated samples with and without SAM as shown in Supplementary Figure 7a. We then measured the force applied as a function of displacement in 10 samples using a universal tensile testing machine.

The average value of the force needed to peel off the PVK layer was determined during the stable process of layer detachment, as illustrated in Supplementary Figures 7b and 7c (also shown below). Our findings reveal that peeling off the PVK layer from a SAM substrate requires more force compared to a NiO_x substrate. This indicates that the 2PACz layer indeed enhances the PVK layer's adhesion strength (Supplementary Figure 7d).

We observed the adhesion strength between the PVK/SAM interface to be 14% stronger than that of the PVK/NiO_x interface. Consequently, to reinforce our conclusion that the strong van der Waals force between PVK and SAM enhances the PVK layer's adhesion, we've incorporated this discussion and corresponding figure into the main text and Supporting Information, respectively.

On page 5:

‘To further examine the mechanical adhesion strength of the PVK/SAM and PVK/NiO_x interfaces, we carried out a series of measurements as depicted in Supplementary Figure 7. Our results indicate a greater force is required to detach the PVK layer from the SAM substrate as compared to the NiO_x substrate. This finding suggests that the 2PACz layer significantly bolsters the adhesion strength of the PVK layer. Additionally, we observed that the average adhesion strength at the PVK/SAM interface is approximately 14% stronger than at the PVK/NiO_x interface.’

In the Methods section:

‘The tensile test was based on a tensile machine developed by Dongriyiqi (DR-509AQ). The samples were prepared with a structure of Glass (15×15 mm)/ITO/NiO_x/SAM/PVK. Then one end of a 3M scotch tape (width of 12.7 mm) was firmly stuck on the surface of PVK. The other end of the tape was fixed on the up holder of the tensile machine for pulling up. Then a double-side tape was stuck to the other side of the sample and firmly mounted on the bottom holder. During pulling up of the up holder, the force-displacement relationship was detected by highly sensitive force and spatial sensors, respectively, and recorded by a computer.’

In Supporting Information:

Supplementary Figure 7 | The adhesion strength measurement between the PVK/SAM and PVK/NiO_x interfaces. **a** The schematic figure of the mechanical adhesion strength test. The force-displacement curves of **(b)** with SAM and **(c)** without SAM layer. The necessary force to peel off the PVK layer is defined as the average value of the plateau of the curve (marked in color background), demonstrating the stable process during the layer peeling off. **d** The statistical data of the calculated adhesive strength of 10 samples (with and without SAM). The adhesion strength is calculated via dividing the necessary force by the width of the tape (12.7 mm). The error bars show the highest and lowest adhesive strength values for samples without and with SAM.

4. In supplementary fig. 5, PL emission wavelength is red shifted from NiO_x to NiO_x/PVK and NiO_x/SAM/PVK. Even if the authors explain that there is a negligible impact on perovskite crystallinity using XRD, this red shift can be explained by more 3D phase in perovskite film on NiO_x/SAM/PVK substrate. The explanation about this wavelength shift would be helpful for the impact on the perovskite layer by different substrates.

Reply:

We thank the reviewer for their comments.

In our previous work, we concluded that different substrates had a negligible effect on the formation of the quasi-2D perovskite phase. This conclusion was based on the observation that the peak positions in the X-ray diffraction (XRD) signals, corresponding to both the 2D and the 3D perovskite regions, remained unchanged. However, upon the reviewer's suggestion, we further examined the XRD signal intensities of the 2D and the 3D phases.

We found a slight increase in the intensity ratio of the 3D to the 2D perovskite peaks as the substrates changed from NiO_x to NiO_x/PVK , and finally to $\text{NiO}_x/\text{SAM}/\text{PVK}$ (Supplementary Figures 9b and 9c, also shown below). This suggested a higher concentration of the 3D phase in the perovskite film formed on the $\text{NiO}_x/\text{SAM}/\text{PVK}$ substrate.

Given the efficient energy and/or charge transfer from the 2D to the 3D phase in quasi-2D perovskite systems, this slight increase in the 3D phase could explain the observed red-shift in the photoluminescence (PL) spectra of perovskite films when the substrates change from NiO_x to NiO_x/PVK , and finally to $\text{NiO}_x/\text{SAM}/\text{PVK}$.

To explain this subtle PL red-shift more clearly, we have revised Supplementary Figure 9 and the associated legend in Supplementary Figure 8 of our manuscript.

In Supplementary Information:

Supplementary Figure 9 | Film quality of $(\text{PEA}/\text{IPA})_2\text{Cs}_{n-1}\text{Pb}_n\text{Br}_{3n+1}$ perovskite on various substrates. **a** Morphologies, Left panel: SEM images; Right panel: AFM images. In both cases, the perovskite film morphologies are similar, suggesting that introducing the 2PACz SAM had a negligible impact on the perovskite morphology. **b** X-ray diffraction patterns. The similar X-ray diffraction peak positions in all cases suggest that different substrates had a negligible impact on forming the quasi-2D perovskite phase. However, the perovskite formed at NiO_x substrate had a poorer crystallinity. **c** The intensity ratio between the 3D and 2D perovskite peaks shown in **b**. An increasing 3D perovskite phase is found when changing the substrates from NiO_x to NiO_x/PVK to $\text{NiO}_x/\text{SAM}/\text{PVK}$.

Supplementary Figure 8 | PL properties of $(\text{PEA/IPA})_2\text{Cs}_{n-1}\text{Pb}_n\text{Br}_{3n+1}$ perovskite films on NiO_x , NiO_x/PVK and $\text{NiO}_x/\text{SAM}/\text{PVK}$ substrates. **a Photos of perovskite films under 365-nm ultraviolet lamp excitation. **b** PL spectra of the corresponding perovskite films. The slight red shift of PL from NiO_x to NiO_x/PVK to $\text{NiO}_x/\text{SAM}/\text{PVK}$ cases is due to the increasing 3D perovskite ratio in quasi-2D perovskite films, according to the XRD results shown in Supplementary Figure 9.**

5. In the introduction, the new record red LED based on quasi-2D perovskites with over 26% EQE should be mentioned (Nature Communications 2023, 14, 397).

Reply:

We agree that important studies should be involved to improve the scientific quality of this manuscript. We therefore, added the aforementioned reference in the revised manuscript.

On page 2:

‘These properties offer much room to improve perovskite light-emitting diodes (PeLEDs) from the material engineering level, leading to > 20% external quantum efficiencies (EQEs) for infrared, red, and green devices, as well as > 17% EQEs for blue and > 12% EQEs for white devices¹⁻⁶.’

In References:

6. Wang, K. et al. Suppressing phase disproportionation in quasi-2D perovskite light-emitting diodes. *Nat. Commun.* **14**, 397 (2023).

6. In the abstract and introduction, the authors should state the wavelength (or clarify it is sky-blue rather than pure blue) of the blue LED before claiming the efficiency of 14.5%. In addition, EL spectra should be provided in the main figures to avoid confusion.

Reply:

We thank the reviewer for their comments. We have stated the EL peak position of the blue PeLEDs in the abstract and introduction sections. To make our description clearer, we have revised our manuscript:

On page 1:

‘We successfully demonstrated blue (emission at 493 nm) and green (emission at 515 nm) PeLEDs with external quantum efficiencies of 14.5% and 26.0%, respectively.’

On page 3:

‘The resulting PeLEDs constitute one of the highest-performing blue PeLEDs [electroluminescence (EL) peak = 493 nm, EQE = 14.5%, luminance = 10392 cd m⁻²] and green PeLEDs (EL peak = 515 nm, EQE = 26.0%, luminance = 83561 cd m⁻²).’

Unfortunately, Fig. 3 in the main text is already very informative and we consider adding more small figures to it difficult. However, we agree with the reviewer that it is important to avoid raising any confusion for the readers. Therefore, we revised the main caption of Fig. 3 in our revised manuscript:

On page 11:

‘Fig. 3 Device performance of blue (EL peak at 493 nm) and green (EL peak at 515 nm) PeLEDs.’

REVIEWERS' COMMENTS

Reviewer #1 (Remarks to the Author):

The authors have addressed the issues raised by the reviewers, and I think the current draft can be publishable.

Reviewer #2 (Remarks to the Author):

I am glad that the authors have fully answered all my questions and corrected all the errors mentioned in my previous comments. Now, this revised manuscript is now logical, detailed, and well-structured. The revision has significantly improved the quality of this manuscript. I recommend this manuscript for publication in Nature Communications as it is.

Reviewer #3 (Remarks to the Author):

The authors have revised the manuscript thoroughly. Therefore, I can recommend publication of this work.